# Heteromeric amyloid filaments of ANXA11 and TDP-43 in FTLD-TDP type C

Diana Arseni[1], Takashi Nonaka[2], Max H. Jacobsen[3], Alexey G. Murzin[1], Laura Cracco[3], Sew Y. Peak-Chew[1], Holly J. Garringer[3], Ito Kawakami[2], Hisaomi Suzuki[4], Misumoto Onaya[4], Yuko Saito[5], Shigeo Murayama[5], Changiz Geula[6], Ruben Vidal[3], Kathy L. Newell[3], Marsel Mesulam[6], Bernardino Ghetti[3], Masato Hasegawa[2] & Benjamin Ryskeldi-Falcon[1✉]

Neurodegenerative diseases are characterized by the abnormal filamentous assembly of specific proteins in the central nervous system[1]. Human genetic studies have established a causal role for protein assembly in neurodegeneration[2]. However, the underlying molecular mechanisms remain largely unknown, which is limiting progress in developing clinical tools for these diseases. Recent advances in cryo-electron microscopy have enabled the structures of the protein filaments to be determined from the brains of patients[1]. All neurodegenerative diseases studied to date have been characterized by the self-assembly of proteins in homomeric amyloid filaments, including that of TAR DNA-binding protein 43 (TDP-43) in amyotrophic lateral sclerosis (ALS) and frontotemporal lobar degeneration with TDP-43 inclusions (FTLD-TDP) types A and B[3,4]. Here we used cryo-electron microscopy to determine filament structures from the brains of individuals with FTLD-TDP type C, one of the most common forms of sporadic FTLD-TDP. Unexpectedly, the structures revealed that a second protein, annexin A11 (ANXA11), co-assembles with TDP-43 in heteromeric amyloid filaments. The ordered filament fold is formed by TDP-43 residues G282/G284–N345 and ANXA11 residues L39–Y74 from their respective low-complexity domains. Regions of TDP-43 and ANXA11 that were previously implicated in protein–protein interactions form an extensive hydrophobic interface at the centre of the filament fold. Immunoblots of the filaments revealed that the majority of ANXA11 exists as an approximately 22 kDa N-terminal fragment lacking the annexin core domain. Immunohistochemistry of brain sections showed the colocalization of ANXA11 and TDP-43 in inclusions, redefining the histopathology of FTLD-TDP type C. This work establishes a central role for ANXA11 in FTLD-TDP type C. The unprecedented formation of heteromeric amyloid filaments in the human brain revises our understanding of amyloid assembly and may be of significance for the pathogenesis of neurodegenerative diseases.

The abnormal filamentous assembly of TDP-43 is the hallmark of multiple neurodegenerative diseases, including FTLD-TDP, ALS and limbic predominant age-related TDP-43 encephalopathy, as well as of inclusion body myopathy[5–8]. Currently, there are no effective means to diagnose or treat these diseases. Although the wild-type protein assembles in the majority of disease cases, pathogenic variants in the gene encoding TDP-43, *TARDBP*, that increase the propensity of the mutated protein to assemble indicate a causal role for assembly in disease[9,10].

Four types of FTLD-TDP, designated A–D, as well as a provisional fifth type E, are distinguished by the distribution of pathological assembled TDP-43 in the brain and are associated with different frontotemporal dementias (FTDs)[11,12]. In FTLD-TDP type C, neocortical-assembled TDP-43 is predominantly distributed in elongated inclusions within dystrophic neurites, which are somewhat more abundant in the superficial cortical layers[11]. This contrasts with the other types of FTLD-TDP, where assembled TDP-43 is mainly found in inclusions within neuronal soma. Compact neuronal cytoplasmic inclusions (NCIs) of assembled TDP-43 are also present in the hippocampal dentate gyrus and striatum in FTLD-TDP type C. FTLD-TDP type C is most frequently associated with semantic variant primary progressive aphasia, which is characterized by selective neurodegeneration of the anterior temporal lobes and impairments to word comprehension[13]. Unlike the other types of FTLD-TDP, for which pathogenic genetic variants account for a large proportion of cases, no such variation has been

[1]MRC Laboratory of Molecular Biology, Cambridge, UK. [2]Department of Brain and Neurosciences, Tokyo Metropolitan Institute of Medical Science, Tokyo, Japan. [3]Department of Pathology and Laboratory Medicine, Indiana University School of Medicine, Indianapolis, IN, USA. [4]Department of Psychiatry, National Hospital Organization Shimofusa Psychiatric Center, Chiba, Japan. [5]Department of Neuropathology, Tokyo Metropolitan Institute for Geriatrics and Gerontology, Tokyo, Japan. [6]Mesulam Center for Cognitive Neurology and Alzheimer's Disease, Feinberg School of Medicine, Northwestern University, Chicago, IL, USA. ✉e-mail: bfalcon@mrc-lmb.cam.ac.uk

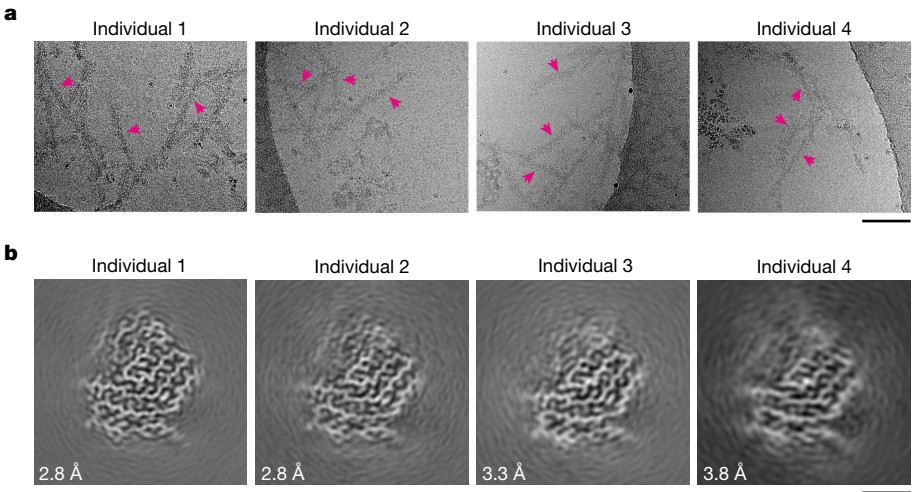

**Fig. 1 | Cryo-EM of filaments from individuals with FTLD-TDP type C.**
**a**, Representative cryo-EM images of filaments extracted from the prefrontal and temporal cortices of four individuals with FTLD-TDP type C. Examples of filaments are indicated with arrows. The filaments were identified by their width of approximately 15 nm, helical crossover distance of approximately 50 nm and granular surfaces. Scale bar, 100 nm. **b**, Cryo-EM reconstructions of filaments from four individuals with FTLD-TDP type C, shown as central slices perpendicular to the helical axis. All four reconstructions have the same filament fold. The resolution of each reconstruction is indicated. Scale bar, 2 nm.

associated with FTLD-TDP type C, which limits our understanding of its pathogenesis.

In its native form, TDP-43 is a ubiquitous RNA-binding protein with multiple regulatory roles in RNA processing[14]. It predominantly localizes to ribonucleoprotein (RNP) granules in the nucleus, but also undergoes nucleocytoplasmic shuttling and can be found in cytoplasmic RNP granules[15]. TDP-43 comprises an N-terminal dishevelled and axin (DIX) domain, a nuclear localization signal, tandem RNA recognition motifs and a C-terminal low-complexity domain (LCD). The LCD contains three distinct regions enriched in glycine, hydrophobic residues, and glutamine and asparagine (Q/N-rich region).

In disease, pathological assembled TDP-43 comprises the full-length protein and truncated C-terminal fragments (CTFs), both of which are abnormally ubiquitylated and phosphorylated[5,6]. The assemblies are filamentous[16–20], but bind to amyloidophilic compounds such as thioflavins poorly[21]. Using cryo-electron microscopy (cryo-EM), we previously established that TDP-43 filaments in ALS and FTLD-TDP types A and B are amyloids[3,4].

Amyloid filaments are characterized by highly stable ordered folds containing parallel, in-register intermolecular β-sheets in line with the filament axis[22]. They have been defined as self-assemblies of identical or near-identical protein sequences. In vitro studies have shown that a given protein sequence has the potential to form a vast number of different filament folds[23]. Each disease studied by cryo-EM to date has been characterized by specific filament folds[1]. This suggests that distinct processes lead to the formation of disease-characteristic filament folds in the brains of patients.

The ordered folds of TDP-43 filaments in ALS and FTLD-TDP types A and B comprise the N-terminal half of the TDP-43 LCD, with flanking sequences forming a fuzzy coat. Despite this, they are folded differently between FTLD-TDP type A and the disease continuum of ALS and FTLD-TDP type B. The structures of TDP-43 filaments in other diseases are not known. Here we used cryo-EM to determine the structures of filaments from the brains of individuals with FTLD-TDP type C.

## Extraction of FTLD-TDP type C filaments

We extracted filaments from the prefrontal and temporal cortices of four individuals with FTLD-TDP type C (Extended Data Table 1), according to the method that we previously used for ALS and FTLD-TDP types

A and B[3,4]. Immunohistochemistry of prefrontal cortex brain sections from these individuals confirmed the presence of abundant elongated TDP-43 inclusions within dystrophic neurites, in the absence of abundant inclusions in neuronal soma (Extended Data Fig. 1a), diagnostic of FTLD-TDP type C[11]. Immuno-gold negative-stain electron microscopy (immuno-EM) of the extracts confirmed the presence of TDP-43-immunoreactive filaments (Extended Data Fig. 1b), as previously reported in extracts[24], and observed in situ within dystrophic neurites of individuals with FTLD-TDP type C[17,18]. These filaments were readily identifiable in cryo-EM images of the extracts (Fig. 1a).

In addition to TDP-43 filaments, we observed occasional amyloid-β filaments in the cryo-EM images for all individuals and tau paired helical filaments for individuals 1–3, which were evident based on their distinct widths and helical crossover distances (Extended Data Fig. 2). This is consistent with the presence of sparse amyloid-β plaques and tau tangles in these individuals (Extended Data Table 1). We also observed distinct filaments of transmembrane protein 106B (TMEM106B) for individuals 1–3, but not for individual 4 (Extended Data Fig. 2), consistent with the previously reported age-dependent accumulation of TMEM106B filaments in the human brain[25]. Individuals 1–3 were all 74 years of age or older, whereas individual 4 died at 59 years of age (Extended Data Table 1). The presence of these filaments in the extracts was supported by mass spectrometry, which identified peptides corresponding to their ordered folds (Supplementary Data 1).

## Cryo-EM reveals heteromeric filaments

We collected just over 300,000 cryo-EM images of the filament extracts and used helical reconstruction to determine the structures of the TDP-43-immunoreactive filaments from each of the four individuals independently, achieving resolutions of between 2.8 Å and 3.8 Å (Fig. 1b, Extended Data Table 2 and Extended Data Fig. 3). We found a single filament type with an identical ordered fold among the four individuals, suggesting that this filament fold characterizes FTLD-TDP type C. The fold comprises two discrete complementary protein chains of different length and conformation (Fig. 2a,b). For individual 1, three-dimensional (3D) classification of the largest dataset of cryo-EM filament segments revealed a variable region at the distal end of the longer chain that adopts two alternative conformations (Extended Data Table 2 and Extended Data Fig. 3). This fold, which

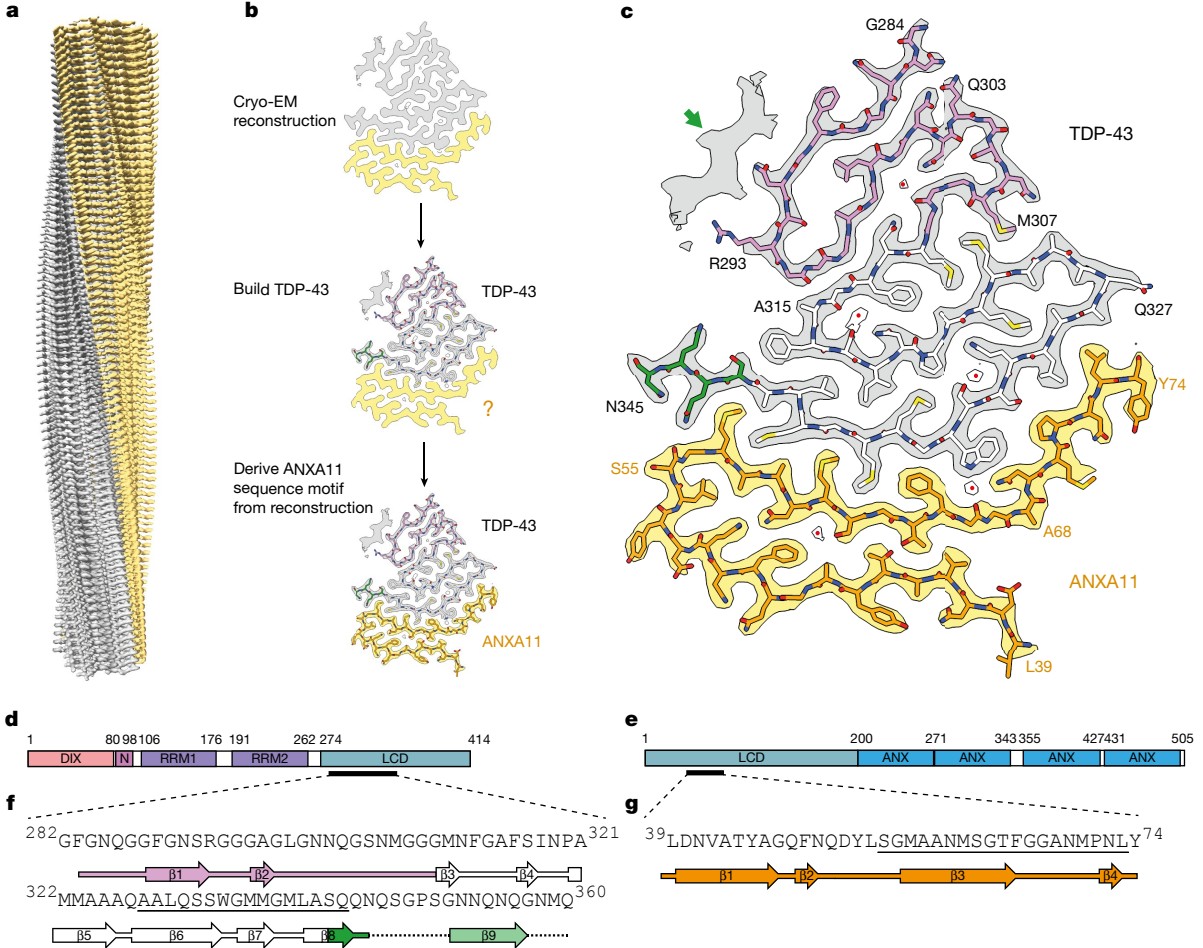

**Fig. 2 | Cryo-EM structure of heteromeric amyloid filaments of ANXA11 and TDP-43 from FTLD-TDP type C. a**, Cryo-EM reconstruction of the left-handed filaments of ANXA11 and TDP-43 from FTLD-TDP type C, shown parallel to the helical axis. **b**, Identification of TDP-43 and ANXA11 chains in the ordered filament fold. ANXA11 was identified by deriving a sequence motif directly from well-resolved amino acid side-chain densities in the cryo-EM reconstruction (see Methods). **c**, Cryo-EM reconstruction and atomic model of the filaments, shown for single TDP-43 and ANXA11 chains perpendicular to the helical axis. The green arrow indicates an isolated peptide consistent with TDP-43 residues N352–G357. Buried ordered solvent is indicated with red dots. **d,e**, Domain

organization of TDP-43 (**d**) and ANXA11 (**e**). ANX, annexin repeat; N, nuclear localization signal; RRM, RNA-recognition motif. The black lines indicate the regions that form the filament fold. **f,g**, Amino acid sequence alignment of the secondary structure elements of the TDP-43 (**f**) and ANXA11 (**g**) chains. The arrows indicate β-strands. The sequences that form the interface between TDP-43 and ANXA11 are underlined. In panels **a–c**, the cryo-EM density for TDP-43 is in grey and ANXA11 is in yellow. In panels **b,c,f,g**, the TDP-43 glycine-rich (G284–G310 in magenta), hydrophobic (M311–S342 in white) and Q/N-rich (Q343–Q345 in green) regions are highlighted. ANXA11 is shown in orange.

resembles a kite (deltoid) in profile, is distinct from the double-spiral fold of ALS and FTLD-TDP type B and the chevron fold of FTLD-TDP type A[3,4] (Extended Data Fig. 4a).

The protein backbone, including peptide group oxygen atoms, and amino acid side chains were unambiguously resolved in our 2.8 Å cryo-EM reconstruction, thereby enabling us to build an accurate atomic model of the filament fold (Fig. 2a–c, Extended Data Table 2 and Extended Data Fig. 3). The filaments have a left-handed helical twist, in contrast with the right-handed filaments of ALS and FTLD-TDP types A and B[3,4]. The longer chain comprises the TDP-43 sequence G282/G284–N345 from its LCD (Fig. 2b–d,f). This is similar to the sequences that form the TDP-43 filament folds of ALS and FTLD-TDP types A and B, but lacks 15 residues from the Q/N-rich region (Q346–Q360; Extended Data Fig. 4b).

Unexpectedly, the shorter second chain could not accommodate any sequences from TDP-43. To identify the protein forming this chain, we derived a sequence motif directly from the well-resolved amino acid side-chain densities of the cryo-EM reconstructions, which returned a single hit when searched against reference proteomes (see Methods;

Fig. 2b). This revealed that the chain belongs to the calcium-dependent phospholipid-binding protein ANXA11 (ref. 26), comprising the sequence L39–Y74 from its N-terminal LCD (Fig. 2b,c,e,g). Among the 12 human annexins, ANXA11 is unique in possessing an LCD[27]. It is enriched in proline residues, apart from the region spanning I37–M70, which lacks proline. The ANXA11 chain comprises almost all of this region, together with four additional residues towards the C terminus. As such, there is only a single proline (P71) in its sequence. We confirmed the presence of both ANXA11 and TDP-43 in the filaments of FTLD-TDP type C using double-labelling immuno-EM (Extended Data Fig. 5). Furthermore, mass spectrometry of the filament extracts detected ANXA11, in addition to TDP-43, with an enrichment of peptides from the ordered filament fold (Supplementary Data 1). All previous structures of neurodegenerative disease-associated filaments from patient brains have been self-assemblies of a single protein within homomeric amyloid filaments[1]. To our knowledge, this is the first evidence that amyloid filaments can be heteromeric in the human brain. This work establishes an unprecedented central role for ANXA11 in FTLD-TDP type C.

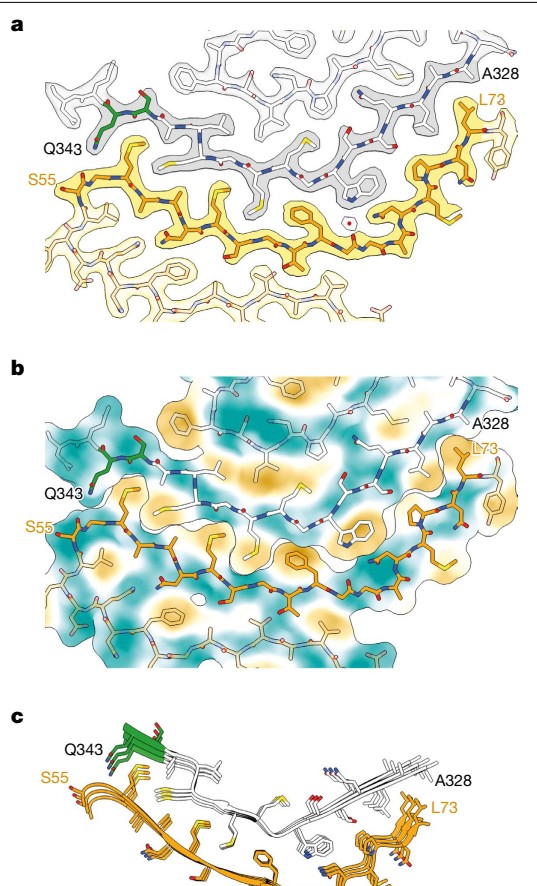

**Fig. 3 | The ANXA11 and TDP-43 interface in heteromeric amyloid filaments from FTLD-TDP type C. a,b**, Overlay of the cryo-EM reconstruction (**a**) and hydrophobicity surface plot (**b**; most hydrophobic in yellow, and least hydrophobic in teal) with the atomic model of the filaments, focused on the interface between ANXA11 and TDP-43 and shown for single ANXA11 and TDP-43 chains perpendicular to the helical axis. The cryo-EM density for TDP-43 is in grey and ANXA11 is in yellow. Buried ordered solvent is indicated with a red dot. **c**, Atomic model of the interface between ANXA11 and TDP-43, shown for three molecular layers perpendicular to the helical axis.

## The ANXA11 and TDP-43 filament fold

One ANXA11 chain and one TDP-43 chain complement each other in the ordered fold of the heteromeric amyloid filaments of FTLD-TDP type C (Fig. 2c and Extended Data Fig. 6a). Residues S55–Y74 from ANXA11 and Q327–N345 from TDP-43 form the most striking interface of the filament fold. This interface comprises two antiparallel layers that associate tightly over their entire lengths, bending in the middle towards the TDP-43 layer (Fig. 3a). Of the 15 amino acid side chains that participate in this interface, 12 (80%) are hydrophobic (Fig. 3b). At one end of the interface, the side chain of TDP-43 residue Q343 forms a hydrogen bond with the main chain of ANXA11 residue S55 (Fig. 3c). In the middle of the interface, there is an ordered solvent molecule that mediates interactions between the polar groups of TDP-43 residue W334 and ANXA11 residues G66 and N69 (Fig. 3a).

The remaining ANXA11 residues, L39–L54, fold back against the opposite side of the ANXA11 interface layer, forming a second layer that extends up to the bend in the interface (Fig. 2c and Extended Data Fig. 6a). The two ANXA11 layers associate through a mixture of non-polar and polar interactions, including sparse hydrogen bonds between Q51 and N60, and between the two threonine residues (T44 and T64; Extended Data Fig. 6b).

The remaining TDP-43 residues, G282/G284–A326, fold inside the bend of the TDP-43 interface layer in a serpentine arrangement (Fig. 2c and Extended Data Fig. 6a). The rest of the TDP-43 hydrophobic region (M311–A326) forms a second layer and half of a third layer. These layers associate through hydrophobic interactions as well as through sparse hydrogen bonding between neutral polar residues (Extended Data Fig. 6b,c). The TDP-43 glycine-rich region (G282/G284–G310) completes the third layer and adds two more layers in the main conformation of the fold and one pleated layer in the alternative conformation (Extended Data Fig. 7a–c). Similar structural variation of the glycine-rich region has previously been observed in the TDP-43 filament fold of FTLD-TDP type A[4].

In the alternative conformation, residues G283–R293 form a compact substructure identical to that of the TDP-43 filament fold of FTLD-TDP type A[4] (Extended Data Fig. 7d). In the latter, the side chain of R293 is buried without charge compensation, whereas in the structure reported here, ordered solvent molecules are present adjacent to the R293 guanidino group, possibly acting as counterions (Extended Data Fig. 7a). As in FTLD-TDP type A, we found partial citrullination and monomethylation of R293 using mass spectrometry of the extracted filaments (Supplementary Data 1). Citrullination would also facilitate the formation of this compact substructure by removing the charge of R293. This suggests that citrullination of R293 may be of broad significance for TDP-43 assembly in disease.

In the main conformation, we observed an isolated density island that probably corresponds to a peptide of approximately six residues engaging in zipper packing with the β-strand formed by F289–R293 in the TDP-43 glycine-rich region (Fig. 2c). This isolated density is consistent with TDP-43 residues N352–G357 and probably represents an extension from N345 at the C terminus of the fold, with the intervening residues (Q346–G351) being disordered or partially absent (Fig. 2f). This density island was not present in the alternative conformation of the glycine-rich region (Extended Data Fig. 7a).

Both the TDP-43 and the ANXA11 chains of the heteromeric filaments are stabilized along the helical axis by hydrogen bonding within intermolecular parallel in-register β-sheets and glutamine/asparagine side-chain ladders, as well as staggered stacking interactions of aromatic side chains, characteristic of amyloid filaments (Extended Data Fig. 6b,d).

## C-terminal truncation of ANXA11

Pathological TDP-43 assemblies contain abnormal CTFs, in addition to the full-length protein[5,6]. To identify the molecular species of ANXA11 in the heteromeric filaments of FTLD-TDP type C, we performed immunoblot analysis of filament extracts (Fig. 4a). This revealed that ANXA11 was predominantly truncated, migrating as an approximately 22-kDa fragment, in addition to a minor population of full-length (56 kDa) ANXA11 (Fig. 4a). The fragment was detected by polyclonal antibodies raised to residues 3–15 and 1–180 of ANXA11, but not to residues 276–505, indicating that it is C-terminally truncated and lacks the annexin core domain (Extended Data Fig. 8a). The ANXA11 N-terminal fragment (NTF) was not detected on immunoblots of filament extracts from individuals with FTLD-TDP types A and B (Fig. 4a), in support of a lack of ANXA11 in the homomeric TDP-43 filament folds of these diseases[3,4]. A faint band for full-length ANXA11 was detected for one individual with FTLD-TDP type B with the highest protein load. Full-length TDP-43 and CTFs of approximately 18–35 kDa phosphorylated at S409 (pS409) and pS410 were detected for all individuals (Fig. 4b), as previously reported[5,6]. These molecular species of ANXA11 and TDP-43 are larger than their sequences in the ordered filament fold (L39–Y74, approximately 3.8 kDa, and G282/284–N345, approximately 6.2 kDa, respectively), indicating that extensive flanking regions extend from the fold to form a fuzzy coat[1]. The ANXA11 NTF has not been reported before and constitutes a new pathological marker of FTLD-TDP type C.

## Redefined pathology of FTLD-TDP type C

Owing to our finding of heteromeric filaments composed of both ANXA11 and TDP-43 in FTLD-TDP type C, we predicted that the TDP-43-immunoreactive inclusions of this disease should also be immunoreactive against ANXA11. Indeed, multiplexed fluorescence immunohistochemistry of prefrontal cortex brain sections from individuals with FTLD-TDP type C showed colocalization between the two proteins in dystrophic neurite inclusions (Fig. 4c and Extended Data Fig. 8b). We did not observe colocalization between ANXA11 and TDP-43-immunoreactive inclusions in prefrontal cortex brain sections from individuals with FTLD-TDP types A and B (Extended Data Fig. 8c,d), in agreement with a lack of ANXA11 in their homotypic TDP-43 filament structures[3,4]. We also observed colocalization between TDP-43 and ANXA11 in the NCIs of the hippocampal dentate gyrus of individuals with FTLD-TDP type C (Extended Data Fig. 9), suggesting that heteromeric amyloid filaments of ANXA11 and TDP-43 are also present in these inclusions. We did not observe inclusions that were immunoreactive against only ANXA11 or only TDP-43, in agreement with our cryo-EM structures in which both proteins co-assemble in heteromeric filaments. ANXA11 immunoreactivity of TDP-43 inclusions redefines the histopathology of FTLD-TDP type C.

## Discussion

The specific proteins that assemble into filaments in neurodegenerative diseases, including TDP-43, tau and α-synuclein, have previously been identified using biochemical fractionation of inclusions from the brains of patients[5,6,28–30], of which they are the major component. Antibodies raised to these proteins have subsequently been used to identify associated diseases. Recent breakthroughs in cryo-EM have enabled the structures of these protein filaments to be determined from the brains of patients[1]. An additional strength of high-resolution cryo-EM is that it can identify novel protein assemblies, as shown for those of TMEM106B in ageing and TAF15 in FTLD[25,31]. Here, using cryo-EM, we found that a second protein, ANXA11, co-assembles with TDP-43 in heteromeric amyloid filaments in FTLD-TDP type C. This reveals a central role for ANXA11 in FTLD-TDP type C. Amyloid filaments were previously defined as self-assemblies of identical or near-identical protein sequences. Indeed, all previously determined structures of neurodegenerative disease-associated filaments from the brains of patients have been homomeric amyloid filaments of a single specific protein[1]. Our discovery of heteromeric amyloid filaments composed of ANXA11 and TDP-43 revises this definition and raises several mechanistic hypotheses.

Pathogenic variants in *ANXA11* have been linked to ALS, inclusion body myopathy and FTD[32–35]. This includes semantic variant primary progressive aphasia[36], which is most commonly associated with FTLD-TDP type C[11]. Inclusions that are immunoreactive against both TDP-43 and ANXA11 have been reported in such cases[34,37,38]. This suggests that pathogenic *ANXA11* variants might promote the co-assembly of ANXA11 and TDP-43 in heteromeric amyloid filaments. Individuals carrying pathogenic variants in genes encoding other LCD-containing proteins, such as hnRNPA1/hnRNPA2B1 and ataxin-2 (refs. 39,40), also exhibit brain inclusions that are immunoreactive against both TDP-43 and the mutated protein. This raises the possibility that TDP-43 may form heteromeric amyloid filaments with these proteins in such cases.

The individuals studied here had wild-type *ANXA11*. This suggests that, in addition to FTLD-TDP type C, subtypes of sporadic ALS and inclusion body myopathy might be characterized by heteromeric amyloid filaments of wild-type ANXA11 and TDP-43. This work should motivate histological examination of additional TDP-43 diseases for ANXA11 pathology. While this article was under review, another study confirmed our finding of colocalization between TDP-43 and ANXA11 in the inclusions of FTLD-TDP type C, as well as the presence of a fragment

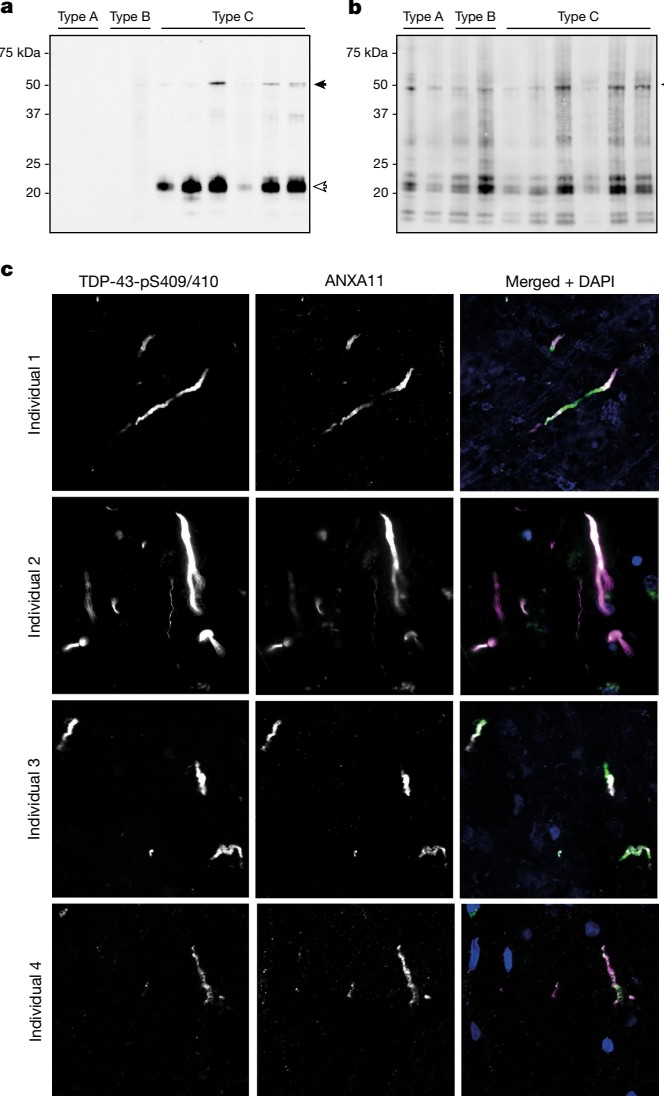

**Fig. 4 | Molecular pathology of ANXA11 in FTLD-TDP type C. a,b,** Immunoblot analysis of filament extracts from the prefrontal cortex of individuals with FTLD-TDP types A (two individuals), B (two individuals) and C (six individuals) using antibodies to N-terminal ANXA11 (**a**; residues 1–180) and TDP-43-pS409/pS410 (**b**). An approximately 22-kDa ANXA11 NTF (white arrow in **a**) and a minor population of full-length ANXA11 (black arrow in **a**) are observed for all individuals with FTLD-TDP type C, but not for individuals with FTLD-TDP types A and B. Full-length TDP-43 (black arrow in **b**) and TDP-43 CTFs (black line in **b**) are observed for all individuals. **c,** Immunohistochemical analysis of prefrontal cortex sections from four individuals with FTLD-TDP type C using antibodies to TDP-43-pS409/pS410 and N-terminal ANXA11 (residues 1–180). Individual images for TDP-43 and ANXA11 are shown in greyscale to facilitate comparison, in addition to a merged image showing TDP-43 (green), ANXA11 (magenta) and DAPI (blue) staining. ANXA11 and TDP-43 colocalize with inclusions. Additional immunolabelling analyses are shown in Extended Data Figs. 8 and 9. Scale bar, 20 μm.

of ANXA11 consistent with the NTF detailed here[41]. Our discovery of the co-assembly of ANXA11 and TDP-43 in heteromeric amyloid filaments explains their colocalization. This other study also reported incomplete colocalization of ANXA11 with TDP-43 inclusions and the presence of an ANXA11 fragment in a small subset of cases of additional TDP-43 proteinopathies. Whether this is also accounted for by the co-assembly of the two proteins in heteromeric filaments remains to be investigated.

The discovery of a second protein in the filaments of FTLD-TDP type C offers new avenues to investigate the currently enigmatic mechanisms of pathological protein assembly in neurodegenerative diseases. The extensive hydrophobic interface between ANXA11 and TDP-43 at the centre of the filament fold strongly suggests that the two proteins co-assemble, rather than forming individual protofilaments. Co-assembly is supported by the absence of homotypic filaments of ANXA11 or TDP-43. ANXA11 and TDP-43 co-exist in axonal RNP granules under physiological conditions[42,43], suggesting that this is where the two proteins might co-assemble. This may explain the distinct distribution of the filaments within neuritic inclusions in FTLD-TDP type C.

The LCD of ANXA11 is required for its association with RNP granules[43], but the underlying interactions are unknown. Our discovery of a pathological interaction between the LCDs of ANXA11 and TDP-43 raises the question of whether this represents the dysregulation of a physiological interaction. The regions of ANXA11 and TDP-43 that interact in the filament fold both form amphipathic α-helices in solution and have been implicated in protein–protein interactions[27,32,44–47]. Possibly, such interactions might precede amyloid co-assembly. Promoting interactions with other binding partners and increasing helical propensity may represent strategies to prevent co-assembly. Future work should focus on producing model systems that recapitulate the heteromeric amyloid filament structure of ANXA11 and TDP-43 to investigate these hypotheses.

We found that ANXA11 in the heteromeric filaments predominantly exists as a previously undescribed approximately 22-kDa NTF lacking the phospholipid-binding annexin core domain. Removal of this domain might facilitate co-assembly with TDP-43 by producing a pool of non-membrane-associated ANXA11. Alternatively, truncation might occur after filament formation, as has been suggested for TDP-43 (ref. 48). Future studies should focus on the molecular mechanisms of ANXA11 truncation. The ANXA11 NTF may also facilitate the search for biomarkers of FTLD-TDP type C, which are currently lacking.

Heteromeric interactions may also be relevant for TDP-43 assembly in other diseases, as well as for the assembly of other disease-associated proteins. Distinct filament folds are found in different diseases[1], but the mechanisms of their formation are unknown. The interface between ANXA11 and TDP-43 in FTLD-TDP type C filaments is incompatible with the TDP-43 folds of ALS and FTLD-TDP types A and B[3,4], suggesting that heteromeric interactions may be one way to influence folds. Isolated peptides associated with the filament folds of homomeric amyloid filaments from the brains of patients have been observed in cryo-EM structures[4,49]. Their short lengths precluded their identification. They may derive from the same protein that makes up the rest of the filament fold, similar to the isolated peptide associated with TDP-43 residues F289–R293 described here. Alternatively, our finding of heteromeric amyloid filaments raises the possibility that these peptides may be derived from different proteins. Their identification may have similar implications for our understanding of assembly mechanisms in disease.

The heteromeric amyloid filament structure of ANXA11 and TDP-43 explains the colocalization of these two proteins in the inclusions of FTLD-TDP type C. However, it is important to make the distinction that colocalization of a given protein with inclusions is not sufficient evidence for heteromeric assembly, as many additional non-assembled proteins are sequestered in inclusions[50–52], and all previously determined structures of pathological assemblies in neurodegenerative diseases have been homotypic amyloid filaments[1,31]. Cryo-EM is currently unique in its ability to conclusively demonstrate that a given protein assembles in disease.

## Conclusion

The co-assembly of ANXA11 and TDP-43 in heteromeric amyloid filaments in FTLD-TDP type C revises our understanding of amyloids as self-assemblies of identical or near-identical sequences. The relevance of this finding to other neurodegenerative diseases needs to be examined. This work establishes a central role for ANXA11 in FTLD-TDP type C. Targeting the co-assembly of ANXA11 and TDP-43 may represent a selective and specific strategy for the diagnosis and treatment of this disease.

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

Alzheimer disease: identification as the microtubule-associated protein tau. *Proc. Natl Acad. Sci. USA* **85**, 4051–4055 (1988).

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

## Methods

### Human tissue samples

We studied fresh-frozen, post-mortem brain tissue from nine neuro-pathologically confirmed cases of FTLD-TDP type C, three cases of FTLD-TDP type A and three cases of FTLD-TDP type B. Neuropathological diagnosis of FTLD-TDP type was made according to the criteria previously described[11]. The clinicopathological details of the four patients with FTLD-TDP type C used for cryo-EM are shown in Extended Data Table 1. All patients with FTLD-TDP type C had wild-type *ANXA11* and clinical presentations of semantic variant primary progressive aphasia. All patients with FTLD-TDP type A carried *GRN* variants associated with FTLD-TDP type A and had clinical presentations of behavioural variant FTD (bvFTD). Two of the patients with FTLD-TDP type B carried hexanucleotide repeat expansions in *C9ORF72* and had clinical presentations of bvFTD. The third patient with FTLD-TDP type B had wild-type *C9ORF72* and a clinical presentation of ALS and bvFTD. Tissue samples were from the Department of Brain and Neurosciences, Tokyo Metropolitan Institute of Medical Science; the Department of Psychiatry, National Hospital Organization Shimofusa Psychiatric Center; the Department of Neuropathology, Tokyo Metropolitan Institute for Geriatrics and Gerontology; and the Brain Library of the Dementia Laboratory at Indiana University School of Medicine. Their use in this study was approved by the ethical review processes at each institution. Informed consent was obtained from the next of kin of the patients.

### Genetic sequencing

Whole-exome sequencing target enrichment used the SureSelectTX human all-exon library (v6; 58-Mb pairs; Agilent) and high-throughput sequencing was carried out using a HiSeq 4,000 (s × 75 bp paired-end configuration; Illumina). Repeat-primed PCR followed by fragment length analyses were performed to screen for hexanucleotide repeat expansions within the *C9ORF72* gene, as previously described[53]. Oligonucleotides designed to amplify the coding exons and corresponding flanking intronic regions of the *ANXA11* gene were used for PCRs using 50 ng of genomic DNA extracted from brain tissue. The amplified products were purified and underwent direct dideoxy sequencing as previously described[54].

### Amyloid filament extraction

Extraction of amyloid filaments from fresh-frozen prefrontal and temporal cortices was performed as previously described[3]. Grey matter was dissected and homogenized using a Polytron (Kinematica) in 40 volumes (v/w) extraction buffer containing 10 mM Tris-HCl (pH 7.5), 0.8 M NaCl, 10% sucrose and 1 mM ethylene glycol-bis(β-aminoethyl ether)-*N*,*N*,*N*′,*N*′-tetra acetic acid (EGTA). A 25% solution of Sarkosyl in water was added to the homogenates to a final concentration of 2% Sarkosyl. Samples were incubated for 1 h at 37 °C with orbital shaking at 200 rpm. Following incubation, samples were centrifuged at 27,000$g$ for 10 min. The resulting supernatants were centrifuged at 166,000$g$ for 20 min. Supernatants were discarded, and pellets containing filaments were resuspended in 6 ml g$^{-1}$ tissue of extraction buffer containing 1% Sarkosyl by sonication for 5 min at 50% amplitude (Qsonica Q700), followed by fourfold dilution with the same buffer and incubation for 30 min at 37 °C with orbital shaking at 200 rpm. Samples were then centrifuged at 17,000$g$ for 5 min and pellets were discarded. The supernatants were further centrifuged at 166,000$g$ for 20 min followed by resuspension in 1 ml g$^{-1}$ tissue of extraction buffer containing 1% Sarkosyl by incubation for 1 h at 37 °C with orbital shaking at 200 rpm. The samples were centrifuged at 166,000$g$ for 20 min and the final pellets were resuspended in 30 μl g$^{-1}$ tissue of 20 mM Tris-HCl (pH 7.4) and 150 mM NaCl by sonication for 5 min at 50% amplitude (Qsonica Q700). One to two grams of tissue was used for each cryo-EM sample. All centrifugation steps were carried out at 25 °C.

### Immunolabelling

For immunohistochemistry, brain hemispheres were fixed with 10% buffered formalin, sectioned and embedded in paraffin. Deparaffinized sections (8 μm thick) were incubated in 10 mM sodium citrate buffer at 105 °C for 10 min and treated with 95% formic acid for 5 min. Sections were washed and blocked with 10% FCS in PBS. For colorimetric immunohistochemistry, sections were incubated overnight with a primary antibody to TDP-43-pS409/pS410 (CAC-TIP-PTD-M01A, CosmoBio; 1:1,000) in blocking buffer. Sections were then washed and incubated with biotinylated secondary antibodies for 2 h. Labelling was detected using an ABC staining kit (Vector) with 3,3′-diaminobenzidine (DAB). Sections were counterstained with haematoxylin. For multiplexed fluorescent immunohistochemistry, sections were incubated overnight with primary antibodies to TDP-43-pS409/pS410 (CAC-TIP-PTD-M01A, CosmoBio; 1:200 or 1:1,000) and ANXA11 residues 1–180 (10479-2-AP, Proteintech; 1:200 or 1:1,000) in blocking buffer. Sections were washed, and fluorescent secondary antibodies conjugated to Alexa Fluor 488, 568 or 594 (A11008, A32723, A11004 and A32740, Thermo) were added for 2 h, followed by washing. Sections were then treated with 0.1% Sudan Black B (Fujifilm Wako) for 10 min and mounted with Vectashield with 4′,6-diamidino-2-phenylindole (DAPI; Vector Laboratories) or ProLong Gold with DAPI (Thermo).

For immunoblotting, filaments extracted from equal amounts of initial grey matter were disassembled and denatured by incubation with lithium dodecyl sulfate sample buffer (Thermo) containing 4% β-mercaptoethanol for 5 min at 95 °C. Samples were then resolved using 12% Bis-Tris gels (Novex) at 200 V for 45 min and transferred onto nitrocellulose membranes. Membranes were blocked in PBS containing 1% BSA and 0.2% Tween for 30 min at 21 °C and incubated with primary antibodies to TDP-43-pS409/pS410 (CAC-TIP-PTD-M01A, CosmoBio; 1:3,000), ANXA11 residues 3–15 (TA302761, OriGene; 1:2,000), ANXA11 residues 1–180 (10479-2-AP, Proteintech; 1:1,000) and ANXA11 residues 276–505 (STJ29559, St John's Laboratory; 1:1,000) at 21 °C for 1 h. Membranes were then washed three times with PBS containing 0.2% Tween, 5 min each wash, and incubated with secondary antibodies conjugated to StarBright Blue 520 (Bio-Rad), DyLight 800 (Cell Signalling Technologies) or horseradish peroxidase (Thermo). Membranes were then washed three times as above, incubated with enhanced chemiluminescence detection reagents (Cytvia) for the horseradish peroxidase-conjugated secondary antibody, and imaged using a ChemiDoc MP (Bio-Rad).

For immuno-EM, filament extracts were deposited onto carbon-coated 300-mesh copper or nickel grids (Nissin EM and Electron Microscopy Sciences, respectively), blocked with 0.1% gelatin in PBS, and incubated with primary antibodies to TDP-43-pS409/pS410 (CAC-TIP-PTD-M01A, CosmoBio; 1:50) and ANXA11 residues 1–180 (10479-2-AP, Proteintech; 1:50) in PBS containing 0.1% gelatin at 21 °C for 3 h or at 4 °C overnight. After washing with 0.1% gelatin in PBS, the grids were incubated with secondary antibodies conjugated to 10-nm or 6-nm gold particles (1:20, Cytodiagnostics, and 1:40, Electron Microscopy Sciences) in PBS containing 0.1% gelatin at 21 °C for 1 h. The grids were then stained with 2% uranyl acetate or NanoVan (Ted Pella). Images were acquired using 80 keV JEOL JEM-1400 and FEI Tecnai Spirit Bio-Twin electron microscopes equipped with CCD cameras.

### Mass spectrometry

Pelleted filaments extracted from 0.1 g of tissue were disassembled by resuspension in 100 μl hexafluoroisopropanol and incubated overnight at 37 °C with shaking at 200 rpm. The samples were then sonicated for 5 min at 50% amplitude (QSonica Q700) and centrifuged at 166,000$g$ for 15 min. The supernatant containing disassembled filaments was collected and dried by vacuum centrifugation (Savant). A solution of 8 M urea in 50 mM ammonium bicarbonate was added to the dried protein samples and then reduced with 5 mM dithiothreitol at 56 °C for

30 min and alkylated with 10 mM iodoacetamide in the dark at room temperature for 30 min. Samples were diluted to 1 M urea with 50 mM ammonium bicarbonate and digested with chymotrypsin (Promega) at 25 °C overnight. The chymotrypsin was inactivated with formic acid to a final concentration of 0.5%. Samples were then centrifuged at 16,000g for 5 min. The resulting supernatants were desalted and fractionated using custom-made C18 stop-and-go-extraction (STAGE) tips (3 M Empore) packed with porous oligo R3 resin (Thermo Scientific). STAGE tips were equilibrated with 80% acetonitrile (MeCN) containing 0.5% formic acid, followed by 0.5% formic acid. Bound peptides were eluted stepwise with increasing MeCN concentrations from 5% to 60% in 10 mM ammonium bicarbonate and partially dried down by vacuum centrifugation (Savant).

Fractionated peptides were analysed by liquid chromatography with tandem mass spectrometry (LC–MS/MS) using a fully automated Ultimate 3000 RSLC nano System (Thermo Scientific). Peptides were trapped with a PepMap100 C18 5-µm 0.3 × 5 mm nano trap column (Thermo Fisher Scientific) and an Aurora Ultimate TS 75 µm × 25 cm × 1.7 µm C18 column (IonOpticks) using a binary gradient consisting of 0.1% formic acid (buffer A) and 80% MeCN in 0.1% formic acid (buffer B) at a flow rate of 300 nl min$^{-1}$. Eluted peptides were introduced directly via a nanoFlex ion source into an a Q Exactive Plus hybrid quadrupole-Orbitrap mass spectrometer (Thermo Scientific). MS1 spectra were acquired at a resolution of 70,000, mass range of 380–1,500 $m/z$, AGC target of $1 \times 10^6$ and MaxIT of 100 ms; this was followed by MS2 acquisitions of the 15 most intense ions with a resolution of 17,500. Normalized collision energy of 27% and isolation window of 1.2 $m/z$ were used. Dynamic exclusion was set for 30 s.

LC–MS/MS data were searched against the human reviewed database (UniProt; downloaded 2019) using the Mascot search engine (Matrix Science v2.4). Database search parameters were set with a precursor tolerance of 10 ppm and a fragment ion mass tolerance of 0.1 Da. A maximum of three missed chymotrypsin cleavages were allowed. Carbamidomethyl cysteine was set as static modification. Arginine citrullination and methylation, and asparagine and glutamine deamination were specified as variable modifications. Scaffold (v4; Proteome Software Inc.) was used to validate MS/MS-based peptide and protein identifications. MS/MS spectra containing arginine citrullination and methylation were manually confirmed using the Scaffold fragmentation tables.

## Cryo-EM

Filament extracts were incubated with 0.4 mg ml$^{-1}$ pronase (Sigma) for 1 h at 21 °C. Samples were centrifuged at 3,000g for 15 s and supernatants were retained. Three microlitres of sample was applied to glow-discharged 1.2/1.3-µm holey carbon-coated 300-mesh gold grids (Quantifoil) and plunge-frozen in liquid ethane using a Vitrobot Mark IV (Thermo Fisher). Images were acquired using a 300 keV Titan Krios microscope (Thermo Fisher) equipped with a K3 detector (Gatan) and a GIF-quantum energy filter (Gatan) operated at a slit width of 20 eV. Aberration-free image shift within the EPU software (Thermo Fisher) was used during image acquisition. Further details are provided in Extended Data Table 2.

## Helical reconstruction

Movie frames were gain-corrected, aligned, dose-weighted and summed using the motion correction program in RELION-4.0 or RELION-5.0 (ref. 55). The motion-corrected micrographs were used to estimate the contrast transfer function (CTF) using CTFFIND-4.1 (ref. 56). All subsequent image processing was performed using helical reconstruction methods in RELION-4.0 or RELION-5.0 (refs. 57,58). Filaments were picked manually. Reference-free 2D classification was performed to remove image coordinates that did not contain filaments. Initial 3D reference models were generated de novo by producing sinograms from 2D class averages as previously described[59]. 3D auto-refinements with optimization of

the helical twist were performed, followed by Bayesian polishing and CTF refinement[55,60]. 3D classification was used to further remove image coordinates that did not contain filaments, as well as to separate segments with alternative conformations. To achieve higher resolutions, 3D classification was also used to remove lower-resolution filament segments consistent with the final reconstructions. 3D auto-refinement, Bayesian polishing and CTF refinement were then repeated. The final reconstructions were sharpened using the standard post-processing procedures in RELION. The overall resolutions were estimated from Fourier shell correlations of 0.143 between the two independently refined half-maps, using phase randomization to correct for convolution effects of a generous, soft-edged solvent mask[61]. Local resolution estimates were obtained using the same phase-randomization procedure, but with a soft spherical mask that was moved over the entire map. Helical symmetry was imposed using the RELION Helix Toolbox. Further details are provided in Extended Data Table 2.

## Atomic model building and refinement

The ANXA11 chain was identified by deriving the 11-residue sequence motif G[NMQ]X[SA][EDRKQN]M[SA][SAG]X[WF][SAG] from the high-resolution cryo-EM reconstructions by inspection of well-resolved densities for amino acid side chains. This motif was then searched against the combined UniProtKB and Swiss-Prot reference proteomes. The search returned ANXA11 as the single hit and enabled residues L39–Y74 of ANXA11 to be built into the cryo-EM reconstruction de novo. We confirmed this result by using the automated machine-learning approach of ModelAngelo[62] to calculate an initial atomic model without supplying a reference sequence, which also returned residues L39–Y74 of ANXA11. The atomic model of the TDP-43 chain and its alternative conformation were built de novo. The complete atomic models were refined in real-space in COOT[63] using the best-resolved maps. Rebuilding using molecular dynamics was carried out in ISOLDE[64]. The models were refined in Fourier space using REFMAC5 (ref. 65), with appropriate symmetry constraints defined using Servalcat[66]. To confirm the absence of overfitting, the model was shaken, refined in Fourier space against the first half map using REFMAC5 and compared with the second half map. Geometry was validated using MolProbity[67]. Molecular graphics and analyses were performed in ChimeraX[68]. Model statistics are provided in Extended Data Table 2.

## Reporting summary

Further information on research design is available in the Nature Portfolio Reporting Summary linked to this article.

## Data availability

Whole-exome data have been deposited to the National Institute on Aging Alzheimer's Disease Data Storage Site (NIAGADS) under the accession code NG00107. Mass spectrometry data have been deposited to the Proteomics Identifications (PRIDE) database under the accession code PXD055345. Cryo-EM datasets have been deposited to the Electron Microscopy Public Image Archive (EMPIAR) under the accession codes EMPIAR-12248 (individual 1), EMPIAR-12247 (individual 2), EMPIAR-12246 (individual 3) and EMPIAR-12245 (individual 4). Cryo-EM reconstructions have been deposited to the Electron Microscopy Data Bank (EMDB) under the accession codes EMD-50628 and EMD-50621 (individual 1; alternative conformations 1 and 2, respectively), EMD-51358 (individual 2), EMD-51359 (individual 3) and EMD-51360 (individual 4). Atomic models have been deposited to the Protein Data Bank (PDB) under the accession codes 9FOR and 9FOF (alternative conformations 1 and 2, respectively).

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

**Acknowledgements** We thank the individuals and their families for donating brain tissue; the Brain Library of the Dementia Laboratory at Indiana University School of Medicine for supplying tissue from individuals 2, 5 and 6 with FTLD-TDP type C, individuals 1 and 2 with FTLD-TDP type A and individuals 1 and 2 with FTLD-TDP type B; the Alzheimer's Disease Research Center at the Mesulam Center for Cognitive Neurology and Alzheimer's Disease, Feinberg School of Medicine, Northwestern University for supplying tissue from individuals 7–9 with FTLD-TDP type C; the Indiana University School of Medicine Center for Electron Microscopy (iCEM) for support with immuno-EM; the Indiana University School of Medicine Center for Medical Genomics for next-generation DNA sequencing; E. Doud for help with mass spectrometry; the staff at the MRC Laboratory of Molecular Biology Electron Microscopy Facility for access to and support with cryo-EM; the staff at the MRC Laboratory of Molecular Biology Scientific Computing Facility for access to and support with computing; and T. Behr, A. Bertolotti, R. Chen, R. A. Crowther, S. W. Davies, A. Giblin, M. Goedert, S. H. W. Scheres, S. Tetter and N. Varghese for discussions. This work was supported by the Medical Research Council, as part of UK Research and Innovation (MC_UP_1201/25 to B.R.-F.); the Hans Und Ilse Breuer Stiftung (to B.R.-F.); the US National Institutes of Health (R01NS137469 to K.L.N., L.C. and B.R.-F.; P30AG072977, R01AG077444, R01DC008552, P30AG13854, R01AG056258 and R01NS085770 to C.G. and M.M.; and R01-NS110437, RF1-AG071177 and R01-AG080001 to R.V. and B.G); the Japan Agency for Medical Research and Development (AMED; JP20dm0207072 to M.H. and JP21wm0425019 to Y.S. and S.M.); the Japan Science and Technology Agency (JST) Core Research for Evolutional Science and Technology (CREST; JPMJCR18H3 to M.H.); the Japan Society for the Promotion of Science (JSPS) KAKENHI (JP22H04923 (CoBiA) to Y.S. and S.M.); the Integrated Research Initiative for Living Well with Dementia (IRIDE) of the Tokyo Metropolitan Institute for Geriatrics and Gerontology IRIDE (to Y.S. and S.M.); and The Leverhulme Trust (ECF-2022-610 to D.A.). For the purpose of open access, the MRC Laboratory of Molecular Biology has applied a CC BY public copyright licence to any Author Accepted Manuscript version arising.

**Author contributions** I.K., H.S., M.O., Y.S., S.M., C.G., K.L.N., M.M., B.G. and M.H. identified individuals and performed neuropathological examinations. T.N., M.H.J., B.G. and M.H. performed the immunohistochemistry. T.N., H.J.G. and R.V. performed the genetic analyses. D.A., T.N., L.C. and M.H. extracted the filaments. D.A. performed the immunoblot analyses. L.C. and M.H. performed the immuno-EM analyses. S.Y.P.-C. performed the mass spectrometry analyses. D.A. collected the cryo-EM data. D.A., A.G.M. and B.R.-F. analysed the cryo-EM data. B.R.-F. supervised the study. All authors contributed to writing the manuscript.

**Competing interests** The authors declare no competing interests.

**Additional information**
**Correspondence and requests for materials** should be addressed to Benjamin Ryskeldi-Falcon.

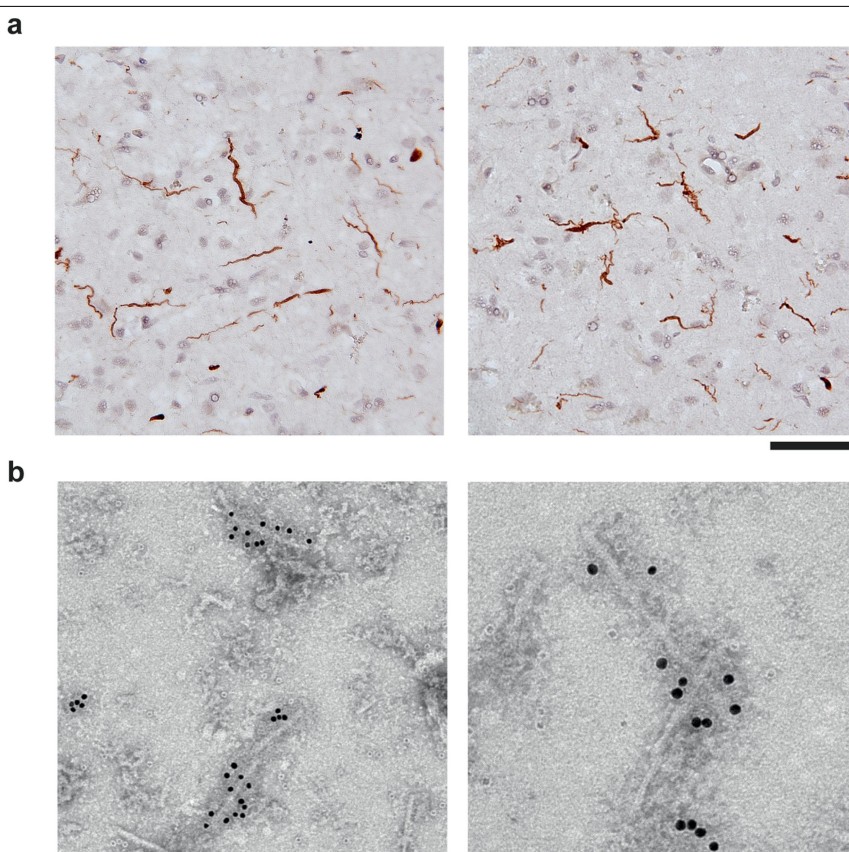

**Extended Data Fig. 1 | Immunohistochemical and immuno-EM analyses of assembled TDP-43 in FTLD-TDP Type C. a**, Immunohistochemical analysis of prefrontal cortex sections from an individual with FTLD-TDP Type C (individual 1) using an antibody against pS409/410 TDP-43 (brown). Sections were counterstained with haematoxylin (blue). Scale bar, 50 μm. **b**, Immuno-EM analysis of filament extracts from the prefrontal cortex of an individual with FTLD-TDP Type C (individual 1) using an antibody against pS409/410 TDP-43 and a 10 nm gold-conjugated secondary antibody (black dots). Scale bars, 100 nm. **a**,**b**, Similar results were obtained for individuals 2–4.

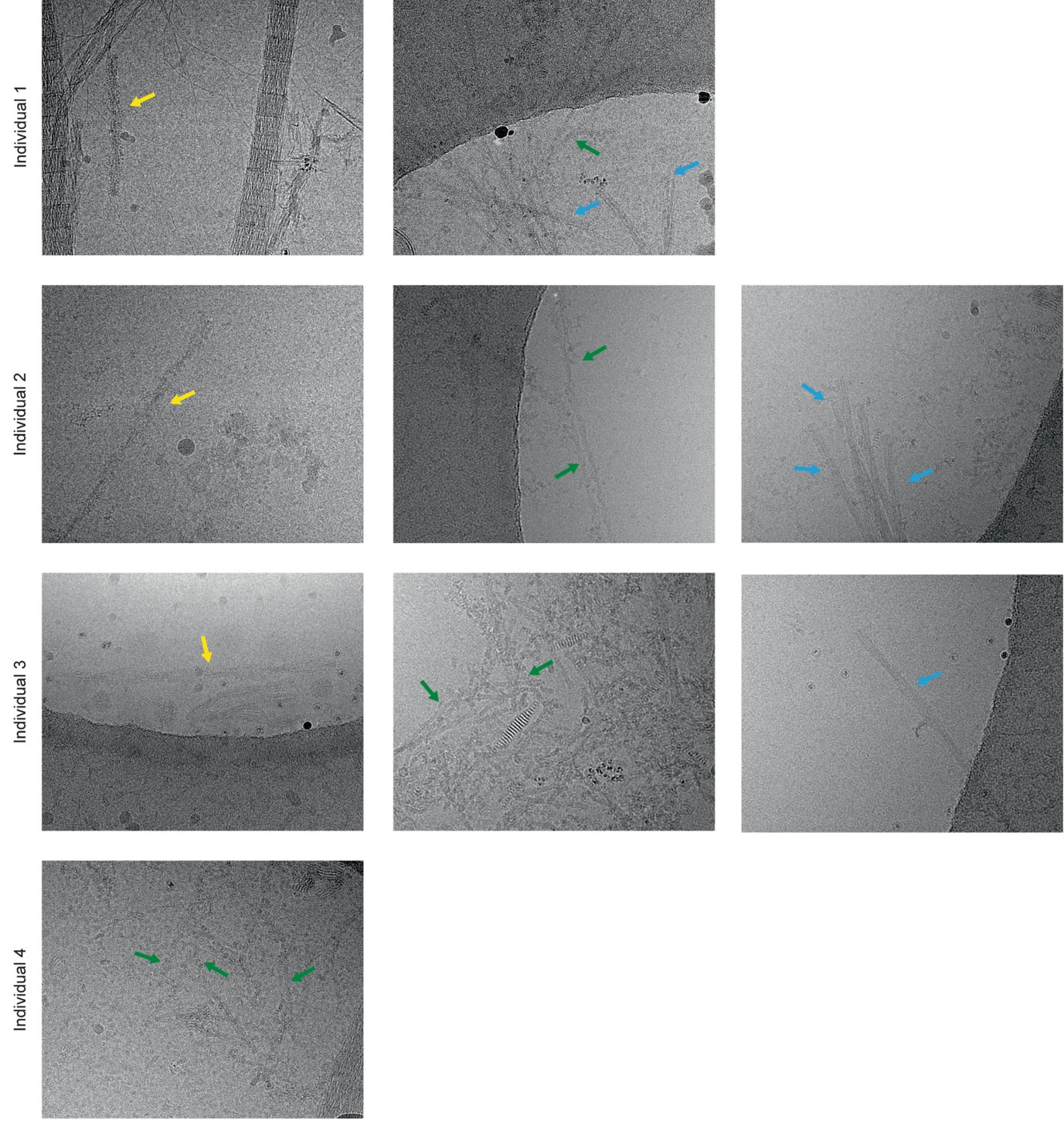

**Extended Data Fig. 2 | Cryo-EM of additional filament types from individuals with FTLD-TDP Type C.** Representative cryo-EM images of tau paired helical filaments (yellow arrows), Aβ filaments (green arrows) and TMEM106B filaments (cyan arrows) in the filament extracts from the prefrontal and temporal cortex of four individuals with FTLD-TDP Type C. Tau paired helical filaments were identified by their width of ~20 nm and helical crossover distance of ~80 nm; Aβ filaments were identified by their width of ~8 nm and helical crossover distance of ~30 nm; and TMEM106B filaments were identified by their widths of ~12 nm (single protofilament) and ~26 nm (double protofilament), helical crossover distances of ~200 nm and smooth surfaces. Scale bar, 100 nm.

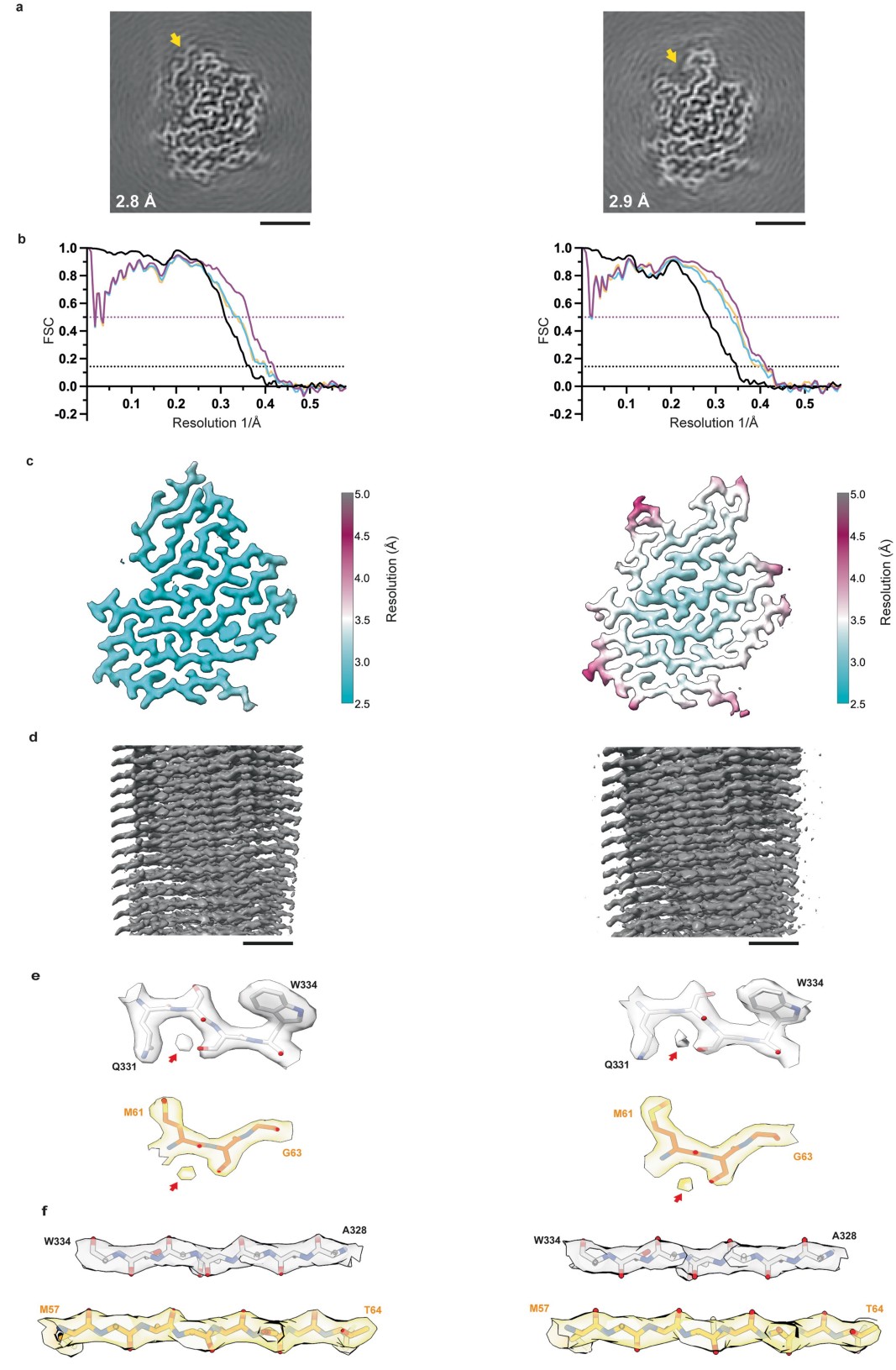

**Extended Data Fig. 3 | Cryo-EM reconstructions and atomic models.**
**a**, Cryo-EM reconstructions of filaments from FTLD-TDP Type C individual 1 with two alternative conformations of the TDP-43 glycine-rich region (indicated with arrows), shown as central slices perpendicular to the helical axis. The resolution of each reconstruction is indicated. Scale bars, 2 nm. **b**, Fourier shell correlation (FSC) curves for the two independently-refined cryo-EM half-maps (black lines); for the refined atomic model against the cryo-EM density map (magenta); for the atomic model shaken and refined using the first half-map against the first half-map (cyan); and for the same atomic model against the second half-map (yellow). FSC thresholds of 0.143 (black dashed line) and 0.5 (magenta dashed line) are shown. **c**, Local resolution estimates for the cryo-EM reconstructions. **d**, Cryo-EM reconstructions viewed along the helical axis. Scale bar, 1 nm. **e**,**f**, Views of the cryo-EM reconstructions and atomic models showing representative densities for ordered solvent (red arrows) (**e**) and main chain oxygen atoms in β-strands (**f**), which reveal the chirality of the map.

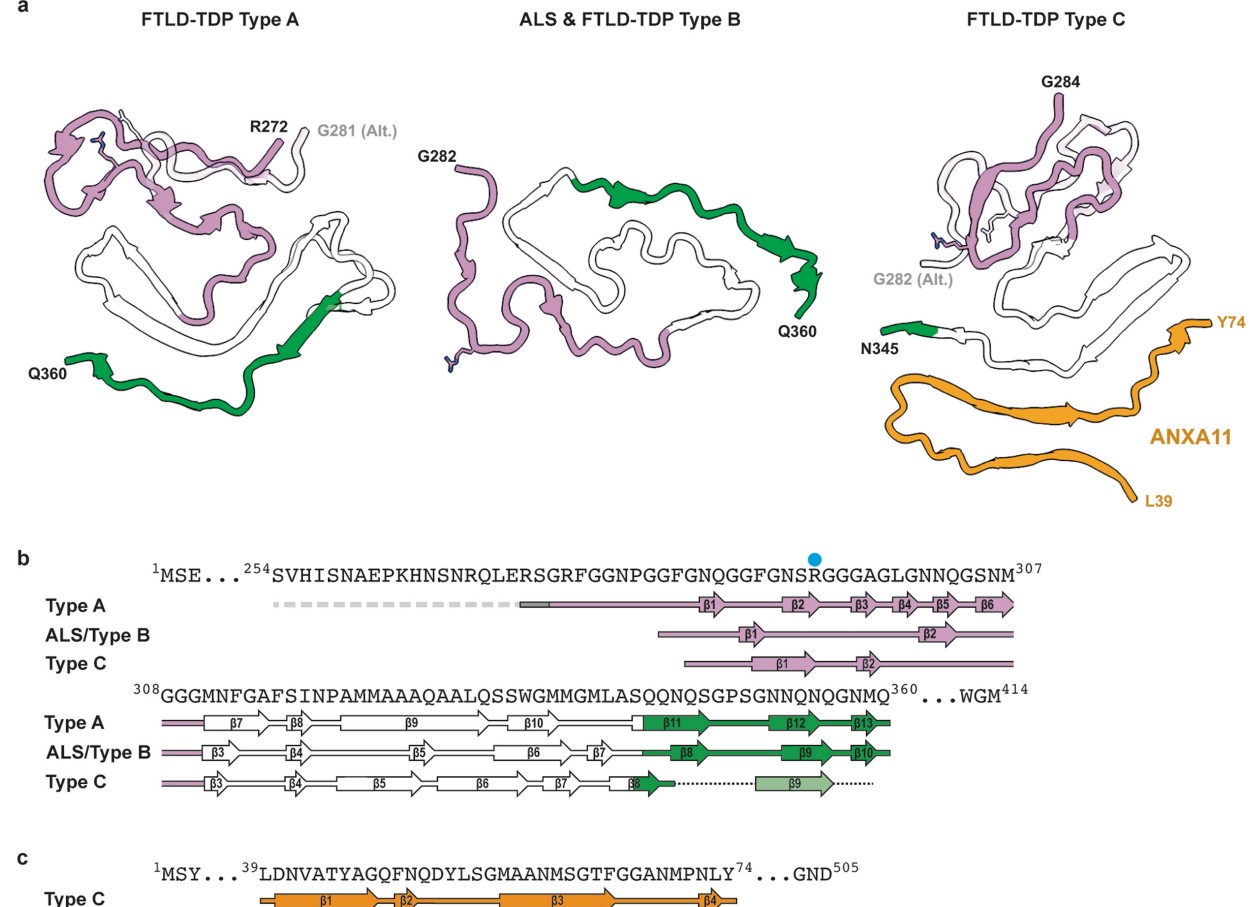

**Extended Data Fig. 4 | Comparison of filament folds in ALS and FTLD-TDP Type A–C. a**, Schematic of the secondary structure elements of the homotypic TDP-43 filament folds of ALS and FTLD-TDP Type A and B, and the heterotypic TDP-43 and ANXA11 fold of FTLD-TDP Type C. Side chains for R293 are shown. Alternative local conformations (Alt.) of the FTLD-TDP Type A and C folds are transparent. **b** and **c**, Amino acid sequence alignment of the secondary structure elements of TDP-43 (**b**) and ANXA11 (**c**) in the filament folds. Arrows indicate β-strands. **a**–**c**, The TDP-43 glycine-rich (G282–G310, magenta), hydrophobic (M311–S342, white) and Q/N-rich (Q343–Q360, green) regions are highlighted. ANXA11 is shown in orange. R293 is indicated with a blue dot.

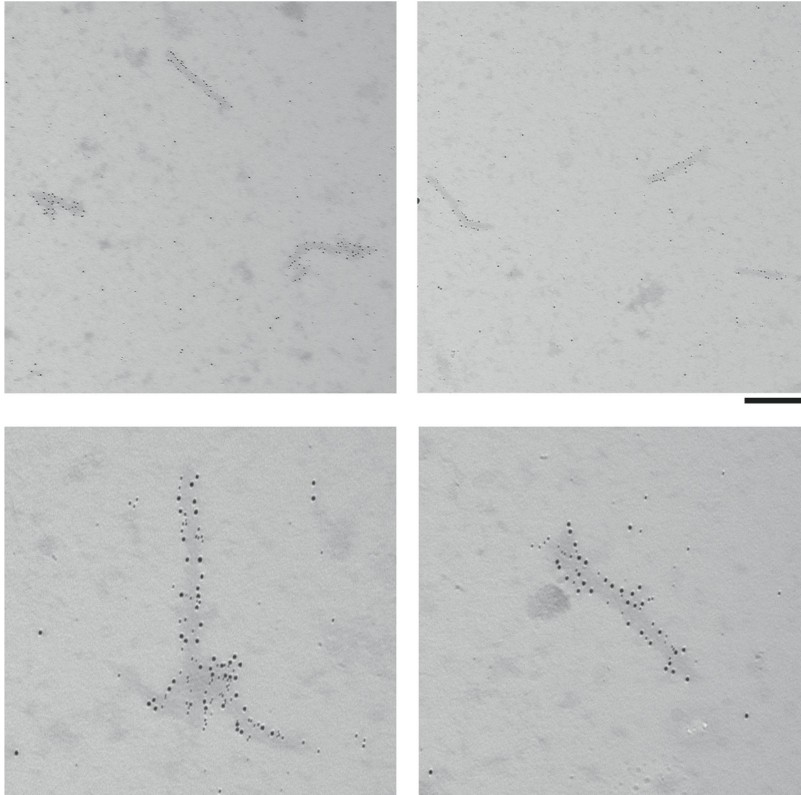

**Extended Data Fig. 5 | Double-labelling immuno-EM of ANXA11 and TDP-43 in FTLD-TDP Type C filaments.** Double labelling immuno-EM analysis of filament extracts from the prefrontal cortex of an individual with FTLD-TDP Type C (individual 2) using antibodies against pS409/410 TDP-43 and N-terminal ANXA11 (residues 1–180) using 10 nm and 6 nm gold-conjugated secondary antibodies (black), respectively. The filaments label for both ANXA11 and TDP-43. Scale bars, 500 nm. Similar results were obtained for another two individuals with FTLD-TDP Type C.

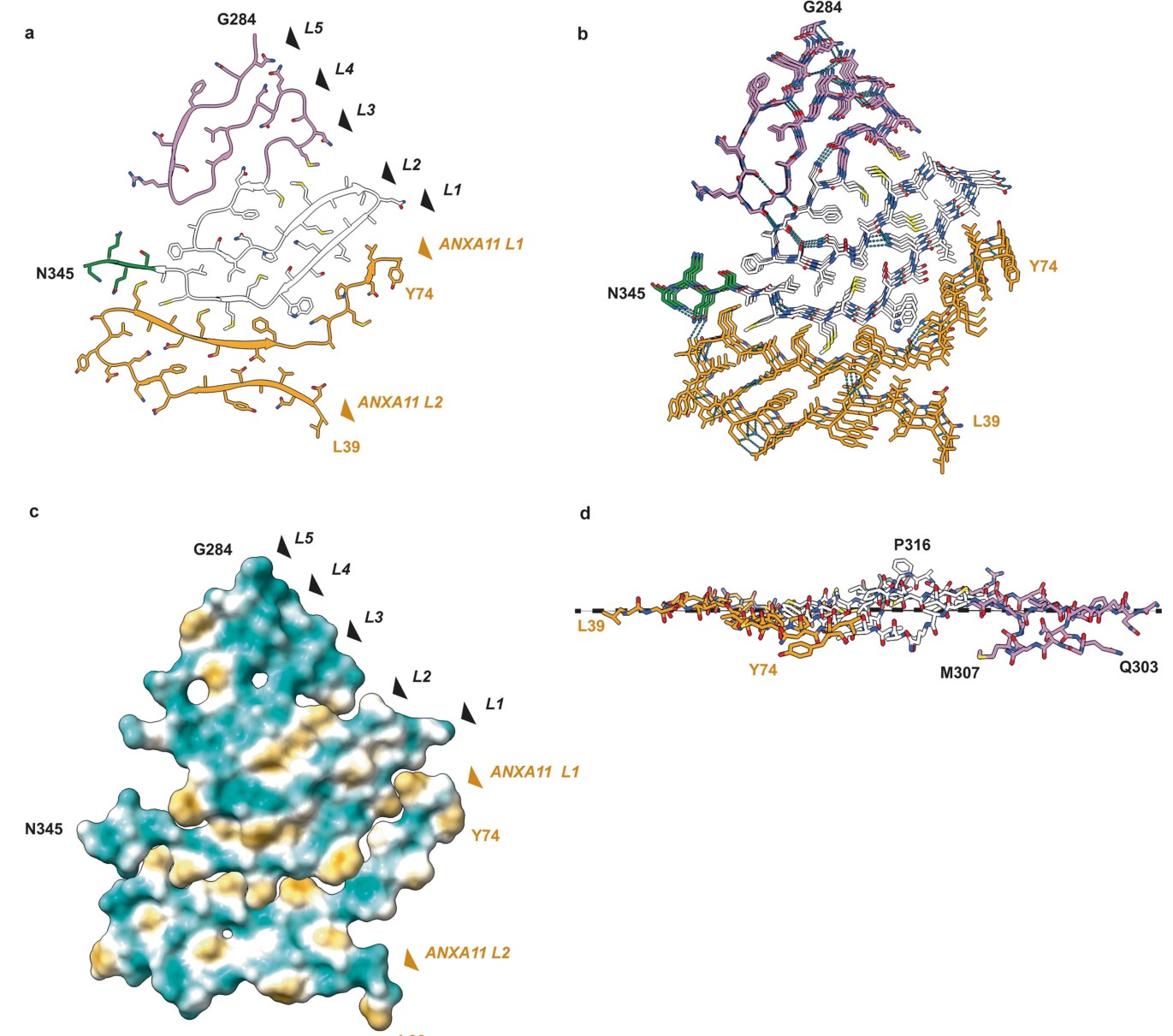

**Extended Data Fig. 6 | The heteromeric filament fold of ANXA11 and TDP-43 from FTLD-TDP Type C. a**, Secondary structure of the heteromeric filament fold of FTLD-TDP Type C, shown for single ANXA11 and TDP-43 molecules perpendicular to the helical axis. **b**, Atomic model of the filament fold depicting hydrogen bonding (dashed cyan lines), shown for three ANXA11 and TDP-43 molecules perpendicular to the helical axis. **c**, Hydrophobicity of the filament fold, from most hydrophilic (teal) to most hydrophobic (yellow), shown for single ANXA11 and TDP-43 molecules perpendicular to the helical axis. **d**, Atomic model of filament fold, shown for single ANXA11 and TDP-43 molecules aligned with the helical axis. **a**, **b** and **d**, The TDP-43 glycine-rich (G284–G310, magenta), hydrophobic (M311–S342, white) and Q/N-rich (Q343–Q345, green) regions are highlighted. ANXA11 is shown in orange. **a** and **c**, The layers of the ANXA11 and TDP-43 chains are indicated with arrows.

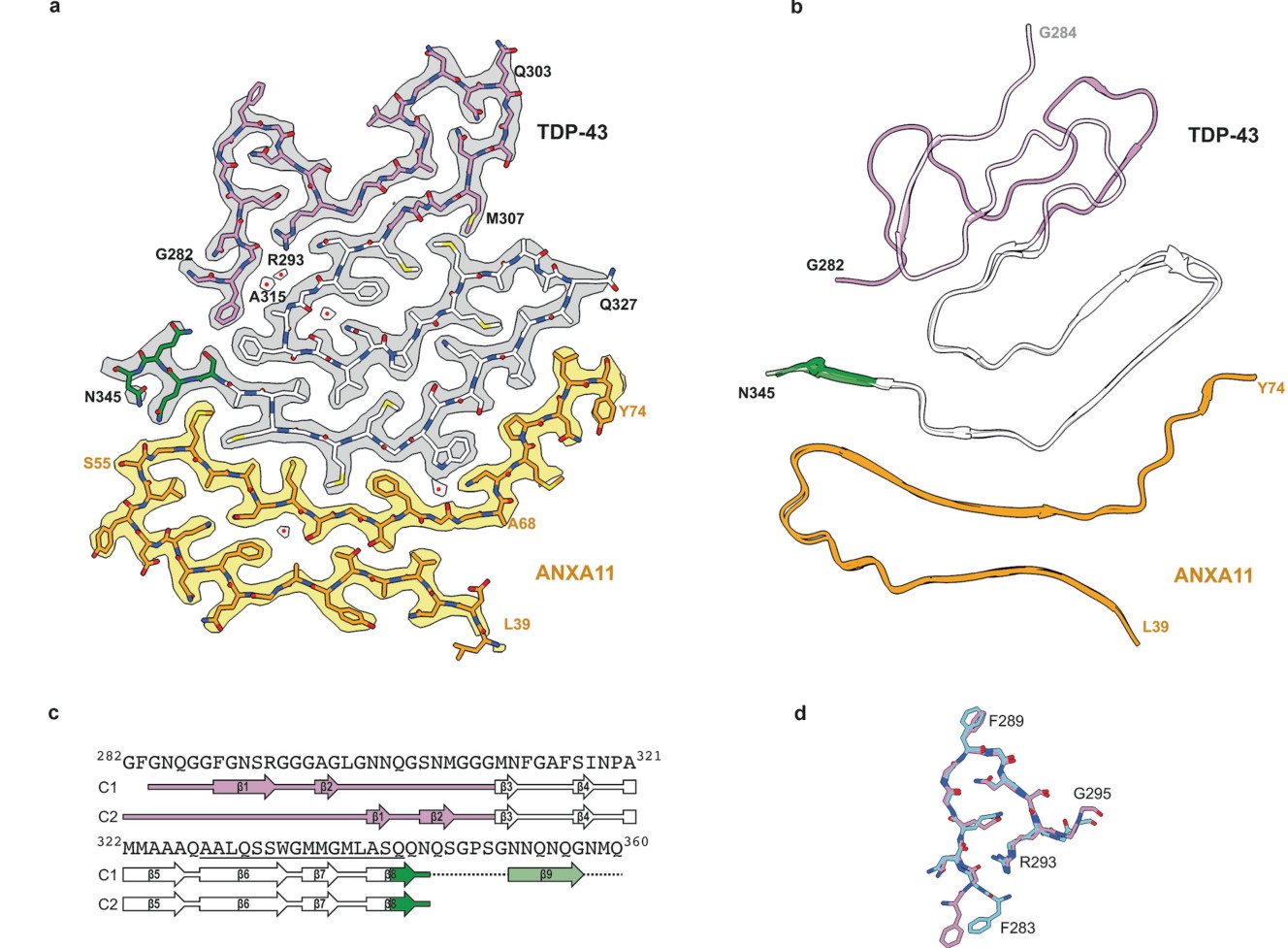

**Extended Data Fig. 7 | Alternative confirmation of the TDP-43 glycine-rich region in heteromeric amyloid filaments from FTLD-TDP Type C. a**, Cryo-EM reconstruction and atomic model of heteromeric amyloid filaments from FTLD-TDP Type C with an alternative conformation of the glycine-rich region, shown for single ANXA11 and TDP-43 molecules perpendicular to the helical axis. Cryo-EM density for TDP-43 is in grey and ANXA11 is in yellow. Buried ordered solvent is indicated with red dots. **b**, Overlay of the atomic models of filaments with the alternative conformation of the glycine-rich region with the main conformation (transparent). **c**, Amino acid sequence alignment of the secondary structure elements of TDP-43 in the filaments. Arrows indicate β-strands. C1, main conformation; C2, alternative conformation. **d**, Alignment of TDP-43 residues N295–R293 from the atomic models of FTLD-TDP Type C (pink) and Type A (cyan) filaments. **a**–**c**, The TDP-43 glycine-rich (G282–G310, magenta), hydrophobic (M311–S342, white) and Q/N-rich (Q343–Q345, green) regions are highlighted. ANXA11 is show in orange.

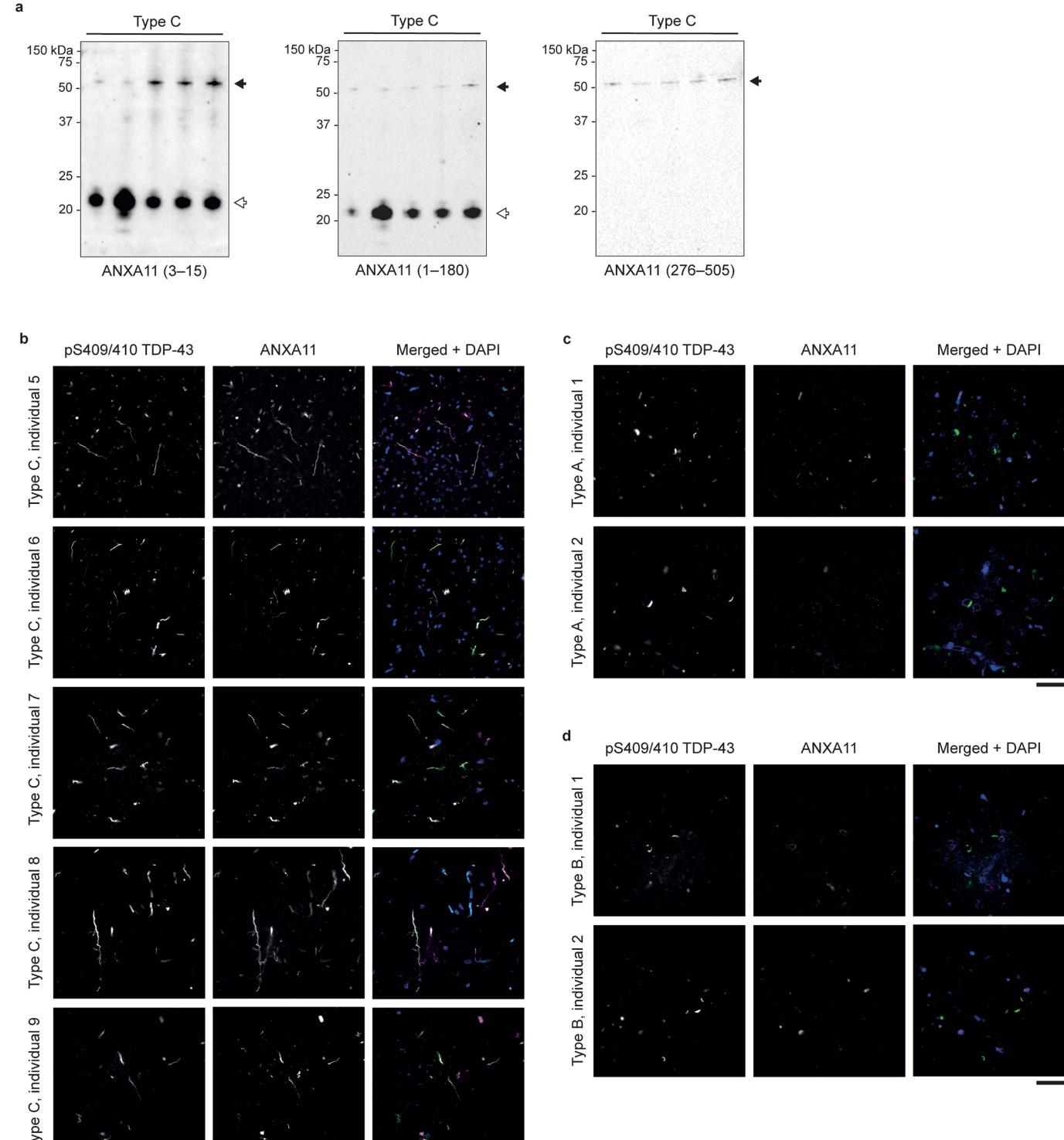

**Extended Data Fig. 8 | Immunoblot and immunohistochemical analysis of ANXA11 and TDP-43 in the prefrontal cortex in FTLD-TDP. a**, Immunoblot analysis of filament extracts from the prefrontal cortex of five individuals with FTLD-TDP Type C using antibodies against residues 3–15, 1–180 or 276–505 of ANXA11. A ~ 22 kDa ANXA11 NTF (white arrow) is observed for the antibodies against residues 3–15 and 1–180, but not 276–505. A minor population of full length ANXA11 (black arrow) is observed for all antibodies. **b**–**d**, Immunohistochemical analysis of prefrontal cortex sections from an additional five individuals with FTLD-TDP Type C (**b**), two with Type A (**c**) and two with Type B (**d**) using antibodies against pS409/410 TDP-43 and N-terminal ANXA11 (residues 1–180). Individual images for TDP-43 and ANXA11 are shown in greyscale to facilitate comparison, in addition to a merged image showing TDP-43 (green), ANXA11 (magenta) and DAPI (blue) staining. ANXA11 and TDP-43 colocalise with inclusions in the individuals with FTLD-TDP Type C, but only TDP-43 co-localises with the inclusions in the individuals with FTLD-TDP Types A and B. Scale bar, 40 μm.

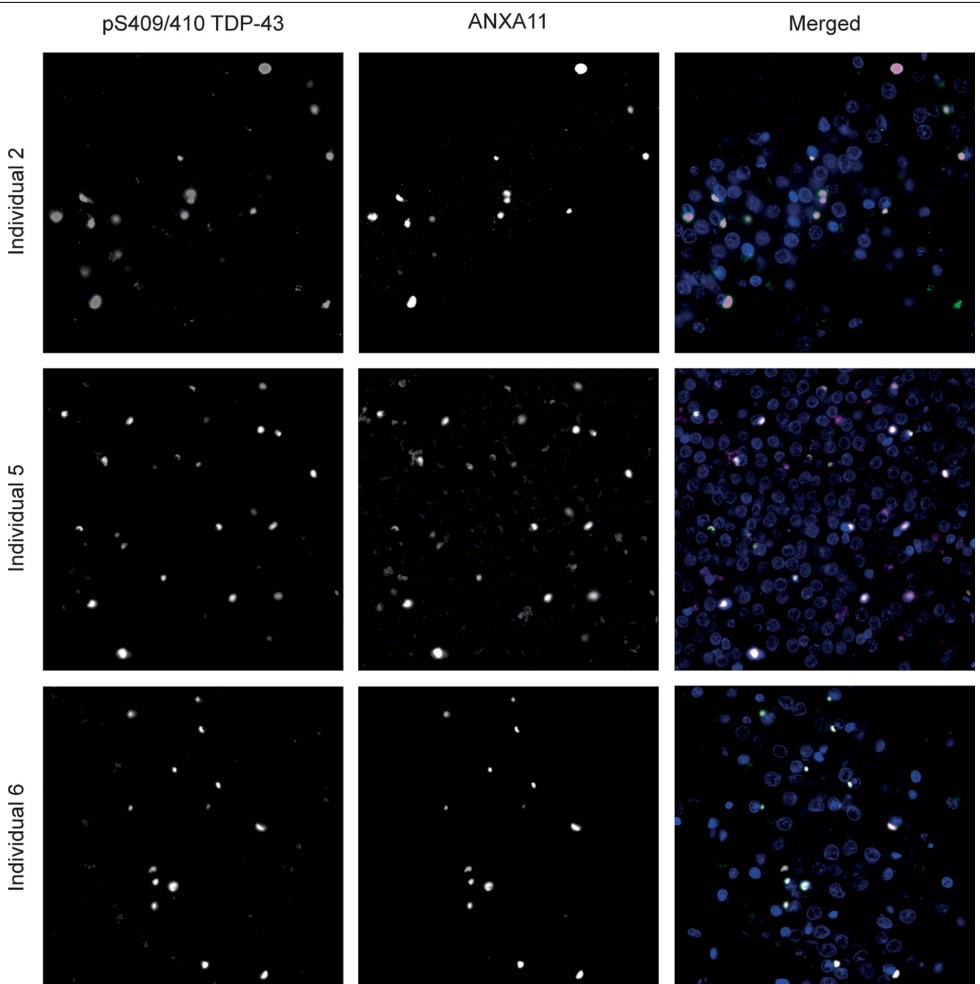

**Extended Data Fig. 9 | Immunohistochemical analysis of ANXA11 and TDP-43 in the hippocampal dentate gyrus in FTLD-TDP Type C.** Immunohistochemical analysis of fascia dentata sections from three individuals with FTLD-TDP Type C using antibodies against pS409/410 TDP-43 and N-terminal ANXA11 (residues 1–180). Individual images for TDP-43 and ANXA11 are shown in greyscale to facilitate comparison, in addition to a merged image showing TDP-43 (green), ANXA11 (magenta) and DAPI (blue) staining. ANXA11 and TDP-43 colocalise with inclusions. Additional immunohistochemical analysis is shown in Fig. 4c and Extended Data Fig. 8. Scale bar, 40 μm.

**Extended Data Table 1 | Clinicopathological details**

|  | Individual 1 | Individual 2 | Individual 3 | Individual 4 |
|---|---|---|---|---|
| Male/ female | F | F | M | M |
| Age (y) | 81 | 74 | 86 | 59 |
| Disease duration (y) | 20 | 11.5 | ND | 12 |
| Clinical diagnosis | svPPA | svPPA | svPPA | svPPA |
| Neuropathological diagnosis | FTLD-TDP Type C | FTLD-TDP Type C | FTLD-TDP Type C | FTLD-TDP Type C |
| Brain regions studied | PFC, TC | PFC | PFC | PFC, TC |
| Thal phase | ND | 1 | 1 | ND |
| Braak stage | 0 | 0 | 1 | 0 |
| CERAD score | ND | 0 | 0 | ND |
| Lewy score | 0 | 0 | 0 | 0 |

F, female; M, male; y, years; svPPA, semantic variant primary progressive aphasia; FTLD-TDP, frontotemporal lobar degeneration with TDP-43 pathology; PFC, prefrontal cortex; TC, temporal cortex; ND, not determined.

**Extended Data Table 2 | Cryo-EM data collection, refinement and validation statistics**

| | Individual 1 (EMPIAR-12248) (EMD-50628 and EMD-50621) (PDB 9FOR and 9FOF) | | Individual 2 (EMPIAR-12247) (EMD-51358) | Individual 3 (EMPIAR-12246) (EMD-51359) | Individual 4 (EMPIAR-12245) (EMD-51360) |
|---|---|---|---|---|---|
| **Data collection** | | | | | |
| Voltage (kV) | 300 | | 300 | 300 | 300 |
| Electron source | XFEG | | XFEG | XFEG | XFEG |
| Detector | K3 | | K3 | K3 | K3 |
| Electron exposure (e–/Å$^2$) | 36 to 57 | | 35 to 77 | 34 to 40 | 45 |
| Defocus range (μm) | -2.2 to -1.0 | | -2.2 to -1.0 | -2.2 to -1.0 | -2.2 to -1.0 |
| Pixel size (Å) | 0.86 | | 0.83 | 0.86 | 0.73 |
| Micrograph number | 148,133 | | 64,122 | 51,501 | 39,379 |
| | | | | | |
| **Data processing** | | | | | |
| Initial particle images (no.) | 619,874 | | 72,366 | 62,112 | 109,728 |
| | Variant 1 | Variant 2 | | | |
| Final particle images (no.) | 18,020 | 10,842 | 8,236 | 10,928 | 9,346 |
| Symmetry imposed | C1 | C1 | C1 | C1 | C1 |
| Helical twist (°) | -1.83 | -1.92 | -1.82 | -1.83 | -1.83 |
| Helical rise (Å) | 4.98 | 4.96 | 4.81 | 4.97 | 4.98 |
| Map resolution (Å) | 2.75 | 2.9 | 2.84 | 3.33 | 3.78 |
| FSC threshold | 0.143 | | 0.143 | 0.143 | 0.143 |
| Map resolution range (Å) | 2.6 to 7.88 | 2.78 to 8.08 | 2.72 to 6.6 | 3.13 to 9.6 | 3.56 to 16.63 |
| | | | | | |
| **Refinement** | | | | | |
| Model resolution (Å) | 2.75 | 2.9 | – | – | – |
| FSC threshold | 0.5 | | – | – | – |
| Map sharpening $B$ factor (Å$^2$) | -47.13 | -50.183 | – | – | – |
| Model composition | | | | | |
| Non-hydrogen atoms | 681 | 696 | – | – | – |
| Protein residues | 98 | 101 | – | – | – |
| Ligands | - | - | – | – | – |
| $B$ factors (Å$^2$) | | | | | |
| TDP-43 | 49.5 | 47.3 | – | – | – |
| ANXA11 | 54.9 | 56.8 | – | – | – |
| R.m.s. deviations | | | | | |
| Bond lengths (Å) | 0.0081 | 0.0089 | – | – | – |
| Bond angles (°) | 1.974 | 1.972 | – | – | – |
| Validation | | | | | |
| MolProbity score | 1.26 | 1.38 | – | – | – |
| Clashscore | 1.23 | 0.75 | – | – | – |
| Favoured rotamers (%) | 98.63 | 98.63 | – | – | – |
| Poor rotamers (%) | 0 | 0 | – | – | – |
| Ramachandran plot | | | | | |
| Favored (%) | 93.62 | 93.75 | – | – | – |
| Allowed (%) | 6.38 | 5.21 | – | – | – |
| Outliers (%) | 0 | 0 | – | – | – |

# Reporting Summary

## Statistics

For all statistical analyses, confirm that the following items are present in the figure legend, table legend, main text, or Methods section.

| n/a | Confirmed | |
|---|---|---|
| ☐ | ☒ | The exact sample size (*n*) for each experimental group/condition, given as a discrete number and unit of measurement |
| ☐ | ☒ | A statement on whether measurements were taken from distinct samples or whether the same sample was measured repeatedly |
| ☒ | ☐ | The statistical test(s) used AND whether they are one- or two-sided<br>*Only common tests should be described solely by name; describe more complex techniques in the Methods section.* |
| ☒ | ☐ | A description of all covariates tested |
| ☒ | ☐ | A description of any assumptions or corrections, such as tests of normality and adjustment for multiple comparisons |
| ☐ | ☒ | A full description of the statistical parameters including central tendency (e.g. means) or other basic estimates (e.g. regression coefficient) AND variation (e.g. standard deviation) or associated estimates of uncertainty (e.g. confidence intervals) |
| ☒ | ☐ | For null hypothesis testing, the test statistic (e.g. *F*, *t*, *r*) with confidence intervals, effect sizes, degrees of freedom and *P* value noted<br>*Give P values as exact values whenever suitable.* |
| ☐ | ☒ | For Bayesian analysis, information on the choice of priors and Markov chain Monte Carlo settings |
| ☒ | ☐ | For hierarchical and complex designs, identification of the appropriate level for tests and full reporting of outcomes |
| ☒ | ☐ | Estimates of effect sizes (e.g. Cohen's *d*, Pearson's *r*), indicating how they were calculated |

*Our web collection on statistics for biologists contains articles on many of the points above.*

## Software and code

Policy information about availability of computer code

| Data collection | EPU 2.14 |
|---|---|
| Data analysis | Mascot 2.4, Scaffold 4.0, RELION 4.0 and 5.0, CTFFIND 4.1, ModelAngelo 1.0, COOT 0.9.8.2, ISOLDE 1.5, REFMAC 5.8.0387, Servalcat 0.3.0, ChimeraX 1.5, MolProbity 4.5.2. |

For manuscripts utilizing custom algorithms or software that are central to the research but not yet described in published literature, software must be made available to editors and reviewers. We strongly encourage code deposition in a community repository (e.g. GitHub). See the Nature Portfolio guidelines for submitting code & software for further information.

## Data

Policy information about availability of data

All manuscripts must include a data availability statement. This statement should provide the following information, where applicable:
- Accession codes, unique identifiers, or web links for publicly available datasets
- A description of any restrictions on data availability
- For clinical datasets or third party data, please ensure that the statement adheres to our policy

Whole-exome data have been deposited to the National Institute on Ageing Alzheimer's Disease Data Storage Site (NIAGADS) under accession code NG00107. Mass spectrometry data has been deposited to the Proteomics Identifications (PRIDE) database under accession code PXD055345. Cryo-EM datasets have been deposited to the Electron Microscopy Public Image Archive (EMPIAR) under accession codes EMPIAR-12248 (Individual 1), EMPIAR-12247 (Individual 2), EMPIAR-12246

(Individual 3) and EMPIAR-12245 (Individual 4). Cryo-EM reconstructions have been deposited to the Electron Microscopy Data Bank (EMDB) under accession codes EMD-50628 and EMD-50621 (Individual 1, alternative conformations 1 and 2, respectively), EMD-51358 (Individual 2), EMD-51359 (Individual 3), and EMD-51360 (Individual 4). Atomic models have been deposited to the Protein Data Bank (PDB) under accession codes 9FOR and 9FOF (alternative conformations 1 and 2, respectively).

## Research involving human participants, their data, or biological material

Policy information about studies with [human participants or human data](). See also policy information about [sex, gender (identity/presentation), and sexual orientation]() and [race, ethnicity and racism]().

| | |
|---|---|
| Reporting on sex and gender | 2 males and 2 females |
| Reporting on race, ethnicity, or other socially relevant groupings | N/A |
| Population characteristics | See Extended Data Table 1. Between 59 and 86 years-of-age. No neurodegenerative disease-associated genetic variants. Clinical diagnosis of svPPA. Neuropathological diagnosis of FTLD-TDP Type C. |
| Recruitment | Selected based on availability and neuropathological examination. |
| Ethics oversight | Human tissue samples were from the Department of Brain and Neurosciences, Tokyo Metropolitan Institute of Medical Science; the Department of Psychiatry, National Hospital Organization Shimofusa Psychiatric Center; the Department of Neuropathology, Tokyo Metropolitan Institute for Geriatrics and Gerontology; and the Brain Library of the Dementia Laboratory at Indiana University School of Medicine. Their use in this study was approved by the ethical review processes at each institution. Informed consent was obtained from the patients' next of kin. |

Note that full information on the approval of the study protocol must also be provided in the manuscript.

# Field-specific reporting

Please select the one below that is the best fit for your research. If you are not sure, read the appropriate sections before making your selection.

☒ Life sciences  ☐ Behavioural & social sciences  ☐ Ecological, evolutionary & environmental sciences

For a reference copy of the document with all sections, see [nature.com/documents/nr-reporting-summary-flat.pdf]()

# Life sciences study design

All studies must disclose on these points even when the disclosure is negative.

| | |
|---|---|
| Sample size | Frontotemporal cortex from 4 individuals with FTLD-TDP Type C. Samples were chosen based on availability and neuropathological examination. |
| Data exclusions | Pre-established common image classification procedures (Scheres 2012. J. Struc. Biol. 180, 519-530) were employed to select particle images with the highest resolution content in the cryo-EM reconstruction process. Details of the number of selected images are given in Extended Data Table 2. |
| Replication | All attempts at replication were successful. Three independent biological repeats per experiment where representative data are shown, as described in the main text. |
| Randomization | Randomisation was not performed. As the samples were limited by brain availability, randomisation would not have reduced any bias in this study. |
| Blinding | The investigators were not blinded to allocation during experiments and outcome assessment. The perceived risk of detection/performance bias was deemed negligible. |

# Reporting for specific materials, systems and methods

We require information from authors about some types of materials, experimental systems and methods used in many studies. Here, indicate whether each material, system or method listed is relevant to your study. If you are not sure if a list item applies to your research, read the appropriate section before selecting a response.

## Materials & experimental systems

| n/a | Involved in the study |
|-----|------------------------|
| ☐ ☒ | Antibodies |
| ☒ ☐ | Eukaryotic cell lines |
| ☒ ☐ | Palaeontology and archaeology |
| ☒ ☐ | Animals and other organisms |
| ☒ ☐ | Clinical data |
| ☒ ☐ | Dual use research of concern |
| ☒ ☐ | Plants |

## Methods

| n/a | Involved in the study |
|-----|------------------------|
| ☒ ☐ | ChIP-seq |
| ☒ ☐ | Flow cytometry |
| ☒ ☐ | MRI-based neuroimaging |

## Antibodies

| | |
|---|---|
| Antibodies used | The primary antibodies used were anti-phospho S409 and 410 TDP-43 (CosmoBio CAC-TIP-PTD-M01A), anti-ANXA11 residues 3–15 (OriGene TA302761), anti-ANXA11 residues 1–180 (Proteintech 10479-2-AP) and anti-ANXA11 residues 276–505 (St John's Laboratory STJ29559). |
| Validation | Validation of anti-phospho S409 and S410 TDP-43 is presented in the manufacturer's datasheet (Cosmo Bio USA) and in (Inukai et al. 2008. FEBS Lett. 582, 2899-2904). Validation of anti-ANXA11 residues 3–15 is presented in the manufacturer's datasheet (OriGene). Validation of anti-Annexin A11 residues 1–180 is presented in the manufacturer's datasheet (Proteintech) and in (Smith et al. 2017. Sci Transl Med 9, eaad9157). Validation of anti-ANXA11 residues 276–505 is presented in the manufacturer's datasheet (St John's Laboratory). |

## Plants

| | |
|---|---|
| Seed stocks | Report on the source of all seed stocks or other plant material used. If applicable, state the seed stock centre and catalogue number. If plant specimens were collected from the field, describe the collection location, date and sampling procedures. |
| Novel plant genotypes | Describe the methods by which all novel plant genotypes were produced. This includes those generated by transgenic approaches, gene editing, chemical/radiation-based mutagenesis and hybridization. For transgenic lines, describe the transformation method, the number of independent lines analyzed and the generation upon which experiments were performed. For gene-edited lines, describe the editor used, the endogenous sequence targeted for editing, the targeting guide RNA sequence (if applicable) and how the editor was applied. |
| Authentication | Describe any authentication procedures for each seed stock used or novel genotype generated. Describe any experiments used to assess the effect of a mutation and, where applicable, how potential secondary effects (e.g. second site T-DNA insertions, mosiacism, off-target gene editing) were examined. |

