## [Peer Review File · Nature]

Manuscript Title: Heteromeric amyloid filaments of ANXA11 and TDP-43 in FTLD-TDP Type C

Reviewer Comments & Author Rebuttals

Reviewer Reports on the Initial Version:

Referees' comments:

Referee #1 (Remarks to the Author):

In this manuscript by Arseni et al, the authors leverage cryo-EM to make a remarkable discovery of heteromeric amyloid filaments composed of ANXA11 and TDP-43 co-assemblies in patients with frontotemporal lobar degeneration (FTLD) type C. This is a foundational discovery and a major milestone in the neurodegeneration field. It is the first clear demonstration of co-assembly of different proteins into organized filaments in disease. Although others have previously described co-localization of multiple aggregation-prone proteins with immunofluorescence microscopy, the prevailing view is that the underlying structures of these aggregates were separate and juxtaposed, not inter-woven. The cryo EM data shown by Arseni et al is definitive, showing that indeed different proteins can co-assemble together into organized filaments in disease. This discovery will change the way that the field conceptualizes mechanisms of aggregation across a range of proteinopathies, potentially including those outside of the nervous system, and will be of interest to the broad research community.

In fact, and as the authors point out, there are numerous other RNA granule proteins that play causal roles in FTD/ALS spectrum disorders associated with TDP-43 proteinopathy, and like ANXA11 and TDP-43, most of these can also cause familial forms of disease when mutated. Previous studies have hinted that some of these proteins, such as ATXN2, may co-localize with TDP-43, suggestive that at least some of these proteins might also co-assemble with TDP-43. This paper demonstrates a methodological path forward toward conclusively determining which proteins actually co-assemble with others, which I suspect will be vigorously applied by numerous groups following publication of this paper to evaluate for other co-assemblies. Finally, the manuscript hints that in certain contexts, the earliest stages of TDP-43 aggregation may occur during normal physiological associations with other RNA granule proteins such as ANXA11. This further suggests that RNPs may be the birthplace of such aggregates, and will lay the groundwork for future mechanistic studies into this process, which is likely central to disease pathogenesis in FTD/ALS spectrum disorders.

The experiments that underlie the above findings were conducted rigorously, and the major conclusions of the manuscript are valid. I do not have major criticisms that would preclude publication. I do have some minor comments that would further strengthen key components of the paper:

- The authors correctly note that the molecular weight of both TDP-43 and ANXA11 present in insoluble material from FTLD-C cases are larger than the core domains of the filaments that were visualized by cryo-EM, indicating that a “fuzzy coat” of each of these proteins likely surrounds the core in filament assemblies. Since the fibrils seen in light microscopy are likely composed of many, many filaments, it is possible that this fuzzy coat also plays roles in filament assembly. Therefore, it will be crucial to map the exact N- and C-terminal sequences of both ANXA11 and TDP-43 that reside outside of the structured filaments. The authors have provided additional mass spec proteomic data that begins to characterize sequences outside of the core domain itself. However, this important data is relegated to a supplemental table and not described in depth.

They should perform additional searches in their mass spec datasets for peptides with non-tryptic cleavage sites; if such peptides are adjacent to full-length tryptic peptides that they also observed, such non-tryptic cleavages may indicate the boundaries of N- and C-terminal peptides. Even better would be to perform additional mass spec proteomic experiments with additional enzymes (chymotrypsin, lys-C, etc) to definitively map N- and C- terminal sequences for TDP-43 and ANXA11 fragments (such experiments would discriminate between non-specific trypsin cleavages vs true non-tryptic peptides).

At the very least, they should graphically display the location of each of the tryptic mass spec peptides against annotated protein sequences of full-length TDP-43 and ANXA11, with some related annotation each peptide intensity on mass spec.

- While this manuscript was in review, a paper was published by Dr. Eddie Lee’s group showing co-localization of ANXA11 and TDP-43 aggregates in a large series of FTLD-C cases via immunofluorescence microscopy of post-mortem brain (PMID: 38896345). These aggregates almost certainly represent the same filaments described at the structural level by Arseni et al. I view these papers as complementary and timely. The authors should cite the Lee paper and discuss the published pathological findings in the context of their new structural findings. Especially since Lee observes interesting co-pathology of ANXA11 and TDP-43 in a fraction of other disorders, such as LATE and FTLD TDP-43 type-B, and since it appears that in these other cases that ANXA11 and TDP-43 can form filaments independent of each other, as well as co-aggregates. Is it possible that such co-aggregates are structurally distinct from those found in FTLD-C? Can ANXA11 form filaments independent of TDP-43, and if so would one expect the ANXA11 filament structures to be distinct from those that are observed with TDP-43? Are they an intermediate filament stage if that is the case? Such questions are clearly beyond the scope of the current paper, but could be speculated further as part of the discussion section.

- There are a few typos related to the C terminal amino acid of the ANXA11 filament (line 151)

Referee #2 (Remarks to the Author):

Identifying the pathological disease proteins in neurodegenerative diseases has always ushered in new insights into pathogenesis. Landmark studies include discovering alpha-synuclein in PD, tau and amyloid beta in AD, TDP-43 in FTLD-TDP and ALS, SOD1 and FUS in fALS, TAF15 in FTLD-FET, etc. For FTLD there are multiple subtypes (A, B, and C) and it has emerged that for type A and type B TDP-43 filaments form unique folds, which may underly their different clinical presentations. The missing piece to the puzzle remained FTLD-TDP type C. Ben Ryskeldi-Falcon and colleagues have now solved this and in so doing revealed something completely new and unexpected that will immediately change the trajectory of the field. Not only did they find a TDP-43 fold in FTLD type C by Cryo-EM that is distinct from type A and type B, but they found the TDP-43 filaments in type C to be heteromeric filaments made up of TDP-43 and N-terminal fragments of another protein! This is the first demonstration of heteromeric filaments in neurodegenerative disease and, I think, will set off a new hunt for other examples like this across neurodegeneration.

The authors start off the paper by continuing their structural exploration of postmortem CNS tissue from patients in the FTLD-TDP/ALS disease spectrum. Turning to FTLD-TDP Type C, they used their well-established cryo-microscopy (cryo-EM) pipeline to determine the ultrastructure of filaments found in cortical brain extracts from 4 individuals diagnosed with svPPA. Surprisingly, the detected amyloid filaments are formed by co-assemblies of TDP-43 and annexin A11 (ANXA11), a protein previously linked to ALS, FTD and inclusion body myopathy (IBM). Compared to previously resolved folds of TDP-43 in FTLD-TDP Types A and B, the ordered fold in subtype C is formed by residues G282/284–N345 of the low complexity domain (LCD), thus excluding 15 amino acids from the Q/N-rich region. The complementary chain of the fold consists of ANXA11 residues L39-L74, which are likewise found in an LCD. The authors use immunohistochemistry to verify (essentially perfect) co-localization of both proteins in patient inclusions and detect an N-terminal fragment (NTF) of ~22 kDa as the major protein species of ANXA11 in filament extracts. Of note, NTF formation and co-assembly of ANXA11 with TDP-43 appear to be specific for subtype C, as they are not observed in FTLD-TDP Types A and B.

The results of this tour de force study are truly remarkable and outstanding for several reasons. While all amyloids found in neurodegenerative diseases to date are composed of a single protein, this study provides the first evidence that amyloid filaments can be heteromeric, which can certainly be considered a paradigm shift in the field. As the authors point out in the Discussion, LCD-LCD interfaces of other proteins may similarly form amyloid structures (e.g., hnRNPA2B1 or ataxin 2), implying that this discovery may serve as a starting point for a new class of amyloids in neurodegeneration. Thus, the current data reinforce the view that phase transitions of LCD-harboring proteins may indeed play a role in the pathogenesis of ALS/FTD: ANXA11 tethers TDP-43 positive RNA granules to lysosomes for RNA transport along axons. This process involves phase separation mediated by the LCD of ANXA11 (Liao et al., Cell 2019). The observation of TDP-43 and ANXA11 forming co-assemblies, particularly in dystrophic neurites (DNs) in FTLD-TDP Type C, suggests a transition from initially physiological, weak interactions to a pathological amyloid state. Thus, the work presented here will encourage mechanistic studies of whether and how such transitions occur in disease. In addition, it also adds another piece to the concept

of the “strain-like” behavior of TDP-43, as the fold detected here differs from those in FTLD-TDP Type A (chevron fold) and FTLD-TDP Type B/ALS (double spiral fold). There is increasing evidence that different polymorphs of aggregation-prone proteins are associated with distinct neurodegenerative diseases, as has been postulated for tau (Shi et al., Nature 2021) and others.

Overall, the manuscript is well written, conveys a succinct and clear message and is based on high quality experiments. Excitingly, shortly after the authors submitted their manuscript, Eddie Lee and colleagues published a paper showing by immunostaining co-localization of fragments of ANXA11 and TDP-43 in FTLD-C and some other diseases. (Robinson et al., Acta Neuropathol 2024). Because co-localization can mean many different things, the novelty and impact of this new manuscript is the discovery of heteromeric filaments made up of both TDP-43 and ANXA11 fragments. These findings will significantly change our (histological) understanding of FTLD-TDP and bring ANXA11 into the spotlight alongside TDP-43, at least in this subtype. I think that this work is a perfect fit for Nature and will surely be an instant classic. I have some comments and suggestions for the authors to consider.

1. The authors should add discussion of the new paper from Eddie Lee and colleagues and how it compares to their findings and new questions and opportunities that both papers open for the field.
2. P2, L56: The authors should consider mentioning the more recently proposed FTLD-TDP subtype “E” (PMID: 28130640).
3. Are N-terminal ANXA11 fragments able to stimulate TDP-43 aggregation? The authors may consider in vitro experiments to test this directly, though this may be beyond the scope of the current manuscript.
4. Mutations in ANXA11 have been described in ALS and FTD. Do these mutant proteins also form heteromeric filaments with TDP-43?
5. Do the heteromeric filaments between TDP-43 and N-terminal fragments of ANXA11 form on the lysosomes or RNA-granules?
6. P3, L78: Why not call it “aggregated” TDP-43? Assembled TDP-43 can also be found in RNP granules, for example, which is not what the authors are referring to in this paragraph.
7. P3, L104: Extended Data Fig. 1a would benefit from a nuclear counterstain.
8. The finding that TDP-43 residue R293 is citrullinated, as in FTLD-TDP type A, is intriguing. However, it is unclear to me why both solvent molecules and citrullination of R293 are detected and proposed to counteract the charge of this side chain. In other words, why are both motifs found at the same time? Or are they present independently in the different conformations? Can the authors clarify this?
9. Besides citrullination of R293, did the authors also detect monomethylation of this residue as described for FTLD-TDP Type A?
10. Can the authors speculate why alternative conformations as well as citrullination of TDP-43 residue R293 are found in FTLD-TDP subtypes A and C, but not B? Is there a (plausible) structural/histological explanation?
11. Have the authors checked for the presence of ANXA11 NTF beyond FTLD-TDP-43 cases, i.e. in the aged population or pure ALS cases? The work from Eddie Lee and colleagues suggests a broader group of diseases with ANXA11 inclusions.
12. Fig. 4a: How is the protein load in filament extracts assessed?
13. For TDP-43, C-terminal fragments (CTFs) are detected more frequently in disease-affected brain

regions than in the spinal cord of patients with ALS and/or FTD (PMID: 31031584). It would be interesting to analyze whether ANXA11 NTFs are likewise region-specific.

Referee #3 (Remarks to the Author):

This is an excellent well written paper that reports a very important result... that two different proteins can aggregate into the same disease-associated amyloid fibril that can be purified from human brain. I must say that I (and many others) have expected this to be the case, so it's very pleasing to see that this is the case. Not only does the result confirm a new 'design principle' of amyloid fibrils, but also suggests new and interesting possibilities about how tissue tropism and disease may be linked to amyloid structure. Overall a fascinating paper. It is certainly worthy of publication in Nature, and I recommend publication with some minor revisions.

1. L88 – I think this is confusing. My reading of their sentence is contradicted by PHF and SF filaments of tau being found in AD brain. I know what they mean, but a bit more clarity in this section would improve it.
2. L124 - What does 296,660 images mean? Micrographs? This is a rather noteworthy large number that I couldn't see in the ED table – and therefore we don't know how these were distributed across the four datasets/patient extracts?
3. L126 – I would include the range of resolutions, not 'up to the highest one'.
4. L138 – handedness? How was the hand of these fibrils determined? I cannot find any reference to unambiguous determination of hand. ED-Fig2 shows some density, but I'm not convinced (from this figure) this is unambiguous determination of hand rather than a cherry picked part of the map.
5. L230 – does one report establish a hallmark? Perhaps 'marker' might be more appropriate?
6. L273-4 – is this a primer for the next paper that the authors intend to send to Nature? If they have these data I would recommend including them here. The novelty of a second observation might preclude publication in a journal of the highest tier...
7. L281 seems grammatically off – needs some rewording.
8. L316 – is there any evidence for this? Presumably there is an extensive body of immunohistochemistry looking for coincidence of other proteins with these deposits. Is there any indication within this literature that the authors speculation is supported by evidence? It seems to me that the likelihood of such densities belonging to the fuzzy coat of the core components, with their effectively infinite local concentration, is a much more likely scenario.
9. I found the overlays in ED-Fig4a hard work – i.e. not immediately accessible – I wonder whether the authors might rework the colour scheme and/or labelling to make the figure clearer.

A few queries about the data...

10. Their 2.8Å and 2.9Å maps... (I don't know why they use a different number of sig figs to describe

resolution of maps – I would suggest their 2.75Å is 2.8Å and they accept the loss of nominal resolution!)
come from 2.9% and 1.7% of the data. What is in the >95% of the data that is not reported on?

11. And on a related note, does this 95% contain any hint of homomeric fibril (of either protein). I assume the answer is 'no' based on the sections about immunohistochemistry – but did they check – it's not clear for example how good the antibodies are...

12. And did they back check in previous datasets knowing the structure and helical parameters of the heterotypic fibril to see if it was in previous datasets from FTLD types A/B.

13. How did they resolve the heterogeneity obviously in their data (obvious from the fact that less than 5% of a dataset ended up in fibril reconstructions? There is little detail, and it might benefit from a processing pipeline ED figure. For example (L482) – what is the definition of suboptimal in “remove suboptimal segments”?

Author Rebuttals to Initial Comments:

We thank the Referees for their constructive and insightful comments, which we feel have substantially improved the manuscript. Please find our point-by-point responses to the individual comments below, written in blue text.

Referees' comments:

Referee #1 (Remarks to the Author):

In this manuscript by Arseni et al, the authors leverage cryo-EM to make a remarkable discovery of heteromeric amyloid filaments composed of ANXA11 and TDP-43 co-assemblies in patients with frontotemporal lobar degeneration (FTLD) type C. This is a foundational discovery and a major milestone in the neurodegeneration field. It is the first clear demonstration of co-assembly of different proteins into organized filaments in disease. Although others have previously described co-localization of multiple aggregation-prone proteins with immunofluorescence microscopy, the prevailing view is that the underlying structures of these aggregates were separate and juxtaposed, not inter-woven. The cryo EM data shown by Arseni et al is definitive, showing that indeed different proteins can co-assemble together into organized filaments in disease. This discovery will change the way that the field conceptualizes mechanisms of aggregation across a range of proteinopathies, potentially including those outside of the nervous system, and will be of interest to the broad research community.

In fact, and as the authors point out, there are numerous other RNA granule proteins that play causal roles in FTD/ALS spectrum disorders associated with TDP-43 proteinopathy, and like ANXA11 and TDP-43, most of these can also cause familial forms of disease when mutated. Previous studies have hinted that some of these proteins, such as ATXN2, may co-localize with TDP-43, suggestive that at least some of these proteins might also co-assemble with TDP-43. This paper demonstrates a methodological path forward toward conclusively determining which proteins actually co-assemble with others, which I suspect will be vigorously applied by numerous groups following publication of this paper to evaluate for other co-assemblies. Finally, the manuscript hints that in certain contexts, the earliest stages of TDP-43 aggregation may occur during normal physiological associations with other RNA granule proteins such as ANXA11. This further suggests that RNPs may be the birthplace of such aggregates, and will lay the groundwork for future mechanistic studies into this process, which is likely central to disease pathogenesis in FTD/ALS spectrum disorders.

The experiments that underlie the above findings were conducted rigorously, and the major conclusions of the manuscript are valid. I do not have major criticisms that would preclude publication. I do have some minor comments that would further strengthen key components of the paper:

- The authors correctly note that the molecular weight of both TDP-43 and ANXA11 present in insoluble material from FTLD-C cases are larger than the core domains of the filaments that were visualized by cryo-EM, indicating that a “fuzzy coat” of each of these proteins likely surrounds the core in filament assemblies. Since the fibrils seen in light microscopy are likely composed of many, many filaments, it is possible that this fuzzy coat also plays roles in filament assembly. Therefore, it will be crucial to map the exact N- and C-terminal sequences of both ANXA11 and TDP-43 that reside outside of the structured filaments. The authors have provided additional mass spec proteomic data that begins to characterize sequences outside of the core domain itself. However, this important data is relegated to a supplemental table and not described in depth.

They should perform additional searches in their mass spec datasets for peptides with non-trypic cleavage sites; if such peptides are adjacent to full-length tryptic peptides that they also observed, such non-trypic cleavages may indicate the boundaries of N- and C-terminal peptides. Even better would be to perform additional mass spec proteomic experiments with additional enzymes (chymotrypsin, lys-C, etc) to definitively map N- and C- terminal sequences for TDP-43 and ANXA11 fragments (such experiments would discriminate between non-specific trypsin cleavages vs true non-trypic peptides).

The detection of the ~22 kDa ANXA11 fragment by a polyclonal antibody raised against residues 1–180 of ANXA11 (Fig. 4), together with the presence of residues L39–Y74 in the filament fold (Fig. 2), indicate that the fragment is C-terminally cleaved somewhere between approximately residues 200 and 250, and might be additionally N-terminally cleaved somewhere before L39.

To narrow down the boundaries of the ANXA11 fragment, we have now carried out additional immunoblot analysis using polyclonal antibodies raised against residues 3–15 and 276–505 of ANXA11. Only the antibody raised against residues 3–15 detected the fragment. This confirms that the fragment is C-terminally cleaved and reveals that it is not N-terminally cleaved. We

have added these new results to a revised Extended Data Fig. 8a and added the following description to the revised manuscript (line 221),

'The fragment was detected by polyclonal antibodies raised against residues 3–15 and 1–180, but not against residues 276–505, indicating that it is C-terminally truncated and lacks the annexin core domain (Extended Data Fig. 8a).'

Regarding our mass spectrometry analysis, we used chymotrypsin instead of trypsin as there are few tryptic cleavage sites in the ANXA11 LCD (only R116 and R191). We did not detect any ANXA11 peptides with non-chymotryptic N-termini, consistent with the lack of N-terminal cleavage indicated by our immunoblot analysis. We detected three peptides with non-chymotryptic C-termini, as shown in Supplementary File 1. Of these, only one peptide, G189–D205, was consistent with the boundary of the ANXA11 fragment defined by immunoblotting, suggesting that D205 could represent the C-terminal cleavage site of the fragment. However, we are mindful of the possibility that this might represent non-specific cleavage by chymotrypsin. We did not detect any semi-tryptic ANXA11 peptides in an additional experiment using trypsin (only two tryptic peptides were detected, G192–K211 and G215–R230). In addition, there is a chance that the peptide terminating at D205 was produced by non-enzymatic aspartyl cleavage during sample preparation for mass spectrometry. Confident assignment of the C-terminal cleavage site will require multiple additional experiments, such as top down proteomics, which are beyond the scope of this study.

Regarding TDP-43, a similar mass spectrometry approach has previously been taken to suggest several N- and C-terminal cleavage sites in ALS and FTL (Nonaka *et al.* 2009. Hum. Mol. Genet. 18,3353–3364; Kametani *et al.* 2016. Sci. Rep. 6,23281). Our mass spectrometry analysis detected several semi-chymotryptic TDP-43 peptides consistent with these studies, as shown in Supplementary File 1.

At the very least, they should graphically display the location of each of the tryptic mass spec peptides against annotated protein sequences of full-length TDP-43 and ANXA11, with some related annotation each peptide intensity on mass spec.

Supplementary File 1 includes peptide counts and annotations for the peptides that map to ordered filament folds. We have now extended these annotations to include protein domains in the revised Supplementary File 1.

- While this manuscript was in review, a paper was published by Dr. Eddie Lee's group showing co-localization of ANXA11 and TDP-43 aggregates in a large series of FTLD-C cases via immunofluorescence microscopy of post-mortem brain (PMID: 38896345). These aggregates almost certainly represent the same filaments described at the structural level by Arseni et al. I view these papers as complementary and timely. The authors should cite the Lee paper and discuss the published pathological findings in the context of their new structural findings. Especially since Lee observes interesting co-pathology of ANXA11 and TDP-43 in a fraction of other disorders, such as LATE and FTLD TDP-43 type-B, and since it appears that in these other cases that ANXA11 and TDP-43 can form filaments independent of each other, as well as co-aggregates. Is it possible that such co-aggregates are structurally distinct from those found in FTLD-C?

We agree that this paper from Eddie Lee's group is complementary and timely. We have now cited this paper and added the following passage to the revised manuscript (line 282),

'While this manuscript was under review, another study confirmed our finding of co-localisation between TDP-43 and ANXA11 in the inclusions of FTLD-TDP Type C, as well as the presence of a fragment of ANXA11 consistent with the NTF detailed here (Robinson et al. 2024. Acta Neuropathol. 147,104). Our discovery of the co-assembly of ANXA11 and TDP-43 in heteromeric amyloid filaments explains their co-localisation. This other study also reported incomplete co-localisation of ANXA11 with TDP-43 inclusions and the presence of an ANXA11 fragment in a small subset of cases of additional TDP-43 proteinopathies. Whether this is also accounted for by the co-assembly of the two proteins in heteromeric filaments remains to be investigated.'

Can ANXA11 form filaments independent of TDP-43, and if so would one expect the ANXA11 filament structures to be distinct from those that are observed with TDP-43? Are they an intermediate filament stage if that is the case? Such questions are clearly beyond the scope of the current paper, but could be speculated further as part of the discussion section.

We did not find any evidence of ANXA11 forming homomeric filaments independently of TDP-43 in FTLD-TDP Type C, as stated in the manuscript (line 296 in the revised version). Hypothetically, if they were to exist in other diseases, their structures would be predicted to be distinct from that of the heteromeric filaments of FTLD-TDP Type C, owing to the extensive

hydrophobic interface with TDP-43 in the heteromeric filaments. The investigation of intermediate stages in filament formation requires the development of model systems that reproduce the final filament structures, as identified in the manuscript (line 309 in the revised version), which is beyond the scope of this study.

- There are a few typos related to the C terminal amino acid of the ANXA11 filament (line 151)

Thank you for spotting this systematic typo, which we have now corrected from L74 to Y74 in the revised manuscript.

Referee #2 (Remarks to the Author):

Identifying the pathological disease proteins in neurodegenerative diseases has always ushered in new insights into pathogenesis. Landmark studies include discovering alpha-synuclein in PD, tau and amyloid beta in AD, TDP-43 in FTLD-TDP and ALS, SOD1 and FUS in fALS, TAF15 in FTLD-FET, etc. For FTLD there are multiple subtypes (A, B, and C) and it has emerged that for type A and type B TDP-43 filaments form unique folds, which may underly their different clinical presentations. The missing piece to the puzzle remained FTLD-TDP type C. Ben Ryskeldi-Falcon and colleagues have now solved this and in so doing revealed something completely new and unexpected that will immediately change the trajectory of the field. Not only did they find a TDP-43 fold in FTLD type C by Cryo-EM that is distinct from type A and type B, but they found the TDP-43 filaments in type C to be heteromeric filaments made up of TDP-43 and N-terminal fragments of another protein! This is the first demonstration of heteromeric filaments in neurodegenerative disease and, I think, will set off a new hunt for other examples like this across neurodegeneration.

The authors start off the paper by continuing their structural exploration of postmortem CNS tissue from patients in the FTLD-TDP/ALS disease spectrum. Turning to FTLD-TDP Type C, they used their well-established cryo-microscopy (cryo-EM) pipeline to determine the ultrastructure of filaments found in cortical brain extracts from 4 individuals diagnosed with svPPA. Surprisingly, the detected amyloid filaments are formed by co-assemblies of TDP-43 and annexin A11 (ANXA11), a protein previously linked to ALS, FTD and inclusion body myopathy (IBM). Compared to previously resolved folds of TDP-43 in FTLD-TDP Types A and B, the ordered fold in subtype C is formed by residues G282/284–N345 of the low complexity domain (LCD), thus excluding 15 amino acids from the Q/N-rich region. The

complementary chain of the fold consists of ANXA11 residues L39-L74, which are likewise found in an LCD. The authors use immunohistochemistry to verify (essentially perfect) co-localization of both proteins in patient inclusions and detect an N-terminal fragment (NTF) of ~22 kDa as the major protein species of ANXA11 in filament extracts. Of note, NTF formation and co-assembly of ANXA11 with TDP-43 appear to be specific for subtype C, as they are not observed in FTLD-TDP Types A and B.

The results of this tour de force study are truly remarkable and outstanding for several reasons. While all amyloids found in neurodegenerative diseases to date are composed of a single protein, this study provides the first evidence that amyloid filaments can be heteromeric, which can certainly be considered a paradigm shift in the field. As the authors point out in the Discussion, LCD-LCD interfaces of other proteins may similarly form amyloid structures (e.g., hnRNPA2B1 or ataxin 2), implying that this discovery may serve as a starting point for a new class of amyloids in neurodegeneration. Thus, the current data reinforce the view that phase transitions of LCD-harboring proteins may indeed play a role in the pathogenesis of ALS/FTD: ANXA11 tethers TDP-43 positive RNA granules to lysosomes for RNA transport along axons. This process involves phase separation mediated by the LCD of ANXA11 (Liao et al., Cell 2019). The observation of TDP-43 and ANXA11 forming co-assemblies, particularly in dystrophic neurites (DNs) in FTLD-TDP Type C, suggests a transition from initially physiological, weak interactions to a pathological amyloid state. Thus, the work presented here will encourage mechanistic studies of whether and how such transitions occur in disease. In addition, it also adds another piece to the concept of the “strain-like” behavior of TDP-43, as the fold detected here differs from those in FTLD-TDP Type A (chevron fold) and FTLD-TDP Type B/ALS (double spiral fold). There is increasing evidence that different polymorphs of aggregation-prone proteins are associated with distinct neurodegenerative diseases, as has been postulated for tau (Shi et al., Nature 2021) and others.

Overall, the manuscript is well written, conveys a succinct and clear message and is based on high quality experiments. Excitingly, shortly after the authors submitted their manuscript, Eddie Lee and colleagues published a paper showing by immunostaining co-localization of fragments of ANXA11 and TDP-43 in FTLD-C and some other diseases. (Robinson et al., Acta Neuropathol 2024). Because co-localization can mean many different things, the novelty and impact of this new manuscript is the discovery of heteromeric filaments made up of both TDP-43 and ANXA11 fragments. These findings will significantly change our (histological)

understanding of FTL-D-TDP and bring ANXA11 into the spotlight alongside TDP-43, at least in this subtype. I think that this work is a perfect fit for Nature and will surely be an instant classic. I have some comments and suggestions for the authors to consider.

1. The authors should add discussion of the new paper from Eddie Lee and colleagues and how it compares to their findings and new questions and opportunities that both papers open for the field.

We agree and have now added this to the revised manuscript (line 282), please see our response to the first Referee (point 2) for details.

2. P2, L56: The authors should consider mentioning the more recently proposed FTL-D-TDP subtype “E” (PMID: 28130640).

We have now amended this sentence to include the provisional Type E and relevant citation in the revised manuscript (line 55) as follows,

'Four types of FTL-D-TDP, designated A–D, as well as a provisional fifth Type E, are distinguished by the distribution of assembled TDP-43 in the brain and are associated with different frontotemporal dementias (FTD) (Mackenzie et al. 2011. Acta Neuropathol. 122,111-113; Lee et al. 2017. Acta Neuropathol. 134,65-78).'

3. Are N-terminal ANXA11 fragments able to stimulate TDP-43 aggregation? The authors may consider in vitro experiments to test this directly, though this may be beyond the scope of the current manuscript.

We raised this hypothesis in the manuscript (line 316 in the revised version) and stated that this was an important line of future research established by this work. This is beyond the scope of the current study, because it requires a model system that recapitulate the heteromeric amyloid filament structure of ANXA11 and TDP-43, which currently does not exist. We also identify the development of such models as an important future research direction (line 309 in the revised manuscript).

4. Mutations in ANXA11 have been described in ALS and FTD. Do these mutant proteins also form heteromeric filaments with TDP-43?

We raised this intriguing hypothesis in the manuscript (line 272 in the revised version). Co-localisation of ANXA11 with TDP-43 inclusions in such cases is consistent with their co-

assembly in heteromeric filaments, but is not sufficient evidence to make this conclusion, which will require filament structures from these cases. This is beyond the scope of this study, but remains an important future research direction opened up by this work.

5. Do the heteromeric filaments between TDP-43 and N-terminal fragments of ANXA11 form on the lysosomes or RNA-granules?

This is certainly a fascinating hypothesis raised by our work. In the manuscript (line 297 in the revised version), we raised the hypothesis that TDP-43 and ANXA11 might co-assemble in RNP granules based on the evidence that they are both found in these compartments. We also identified the development of model systems that recapitulate the heteromeric amyloid filament structure of ANXA11 and TDP-43 as an important future research direction to test these hypotheses and shed light on disease mechanism (line 309 in the revised manuscript).

6. P3, L78: Why not call it “aggregated” TDP-43? Assembled TDP-43 can also be found in RNP granules, for example, which is not what the authors are referring to in this paragraph.

We use the term 'abnormal filamentous assemblies' (line 22 in the revised manuscript). We prefer this as a more accurate description compared to 'aggregates,' while maintaining the distinction from physiological assembly in RNP granules. Subsequently in the manuscript, we used the term 'assemblies' as shorthand for 'abnormal filamentous assemblies.' To avoid misinterpretation, we have changed this shorthand term to 'pathological assemblies' in the first instance of each paragraph in the revised manuscript.

7. P3, L104: Extended Data Fig. 1a would benefit from a nuclear counterstain.

We have now replaced the images in this figure part in the revised manuscript with ones that include a nuclear counterstain.

8. The finding that TDP-43 residue R293 is citrullinated, as in FTLTDP type A, is intriguing. However, it is unclear to me why both solvent molecules and citrullination of R293 are detected and proposed to counteract the charge of this side chain. In other words, why are both motifs found at the same time? Or are they present independently in the different conformations? Can the authors clarify this?

This is explained by the fact that there is a mixture of TDP-43 molecules with either citrullinated or non-citrullinated R293 in the filaments. This is shown by the mass spectrometry analysis in Supplementary File 1, which identified both citrullinated and non-citrullinated

R293. To clarify this, we have amended the following sentence in the revised manuscript (line 197),

*'As in FTLD-TDP Type A, we found **partial** citrullination **and** **monomethylation** of R293 using mass spectrometry of the extracted filaments (Supplementary File 1). **Citrullination** would **also** facilitate the formation of this compact substructure by removing the charge of R293.'*

9. Besides citrullination of R293, did the authors also detect monomethylation of this residue as described for FTLD-TDP Type A?

Yes, we also detected monomethylation of R293 in a subset of the TDP-43 molecules in the filaments, as shown in Supplementary File 1. We have clarified this in the revised manuscript with the sentence discussed in the previous comment 8.

10. Can the authors speculate why alternative conformations as well as citrullination of TDP-43 residue R293 are found in FTLD-TDP subtypes A and C, but not B? Is there a (plausible) structural/histological explanation?

We hypothesise above (point 8) that citrullination of R293 facilitates the formation of a compact substructure of the glycine-rich region, in which the side chain of this residue is buried. We found no evidence of citrullination of R293 in our previous mass spectrometry analysis of TDP-43 filaments from two individuals with ALS and FTLD-TDP Type B (Arseni et al. 2022. Nature 601, 139–143), although we cannot rule out that the mass spectrometry failed to detect this. In the ALS/FTLD-TDP Type B filament fold, the side chain of R293 is fully exposed to the solvent. We also note that in the filament folds of FTLD-TDP Types A and C, the glycine-rich region folds upon itself. In contrast, in the ALS/FTLD-TDP Type B filament fold, the glycine-rich region adopts a more extended conformation to shield the hydrophobic-rich region from the solvent. We speculate that this restricts opportunities for the glycine-rich region to form alternative conformations in this latter fold.

11. Have the authors checked for the presence of ANXA11 NTF beyond FTLD-TDP-43 cases, i.e. in the aged population or pure ALS cases? The work from Eddie Lee and colleagues suggests a broader group of diseases with ANXA11 inclusions.

We did not check for the presence of the ANXA11 NTF beyond FTLD-TDP cases. The paper from Eddie Lee's group shows that an ANXA11 fragment that is consistent with this NTF can be detected in a small proportion of LATE and FTLD-TDP Types A and B cases. We have

included a discussion about the broader significance of this observation in the revised manuscript (line 282).

12. Fig. 4a: How is the protein load in filament extracts assessed?

As stated in the Methods (line 708 of the revised manuscript), filaments extracted from equal amounts of initial grey matter were loaded for the immunoblots shown in Fig. 4a and the revised Extended Data Fig. 8a. This reflects the intrinsic differences in filament abundance among individuals.

13. For TDP-43, C-terminal fragments (CTFs) are detected more frequently in disease-affected brain regions than in the spinal cord of patients with ALS and/or FTD (PMID: 31031584). It would be interesting to analyze whether ANXA11 NTFs are likewise region-specific.

Unlike ALS and FTLN-TDP Type B, FTLN-TDP Type C does not exhibit TDP-43 pathology in the spinal cord. Apart from the prefrontal cortex, TDP-43 pathology is found in the hippocampal dentate gyrus and striatum in FTLN-TDP Type C. We showed in Extended Data Fig. 9 that inclusions in the dentate gyrus are also immunoreactive against N-terminal ANXA11. In addition, the study by Eddie Lee's group also detected the ANXA NTF in the amygdala of FTLN-TDP Type C cases (Robinson et al. 2024. *Acta Neuropathol.* 147,104). These results do not support region-specificity of the ANXA11 NTF.

Referee #3 (Remarks to the Author):

This is an excellent well written paper that reports a very important result... that two different proteins can aggregate into the same disease-associated amyloid fibril that can be purified from human brain. I must say that I (and many others) have expected this to be the case, so it's very pleasing to see that this is the case. Not only does the result confirm a new 'design principle' of amyloid fibrils, but also suggests new and interesting possibilities about how tissue tropism and disease may be linked to amyloid structure. Overall a fascinating paper. It is certainly worthy of publication in Nature, and I recommend publication with some minor revisions.

1. L88 – I think this is confusing. My reading of their sentence is contradicted by PHF and SF filaments of tau being found in AD brain. I know what they mean, but a bit more clarity in this section would improve it.

We agree that the phrase 'a single filament fold' could cause confusion, especially with regards to AD where there are filament folds of both tau and amyloid- β . We have, therefore, changed

this phrase to 'specific filament folds' to highlight the contrast with the non-specific filament folds found *in vitro*, which is the main message of this paragraph.

2. L124 - What does 296,660 images mean? Micrographs? This is a rather noteworthy large number that I couldn't see in the ED table – and therefore we don't know how these were distributed across the four datasets/patient extracts?

Yes, this number refers to micrographs. We have now added the number of micrographs for each individual to a revised Extended Data Table 2 to show how they are distributed. Upon recalculation, the number of micrographs was in fact found to be 303,135, which we have updated in the revised manuscript (line 124).

3. L126 – I would include the range of resolutions, not 'up to the highest one'.

We agree and have now replaced this with the range of resolutions (line 126 in the revised manuscript).

4. L138 – handedness? How was the hand of these fibrils determined? I cannot find any reference to unambiguous determination of hand. ED-Fig2 shows some density, but I'm not convinced (from this figure) this is unambiguous determination of hand rather than a cherry picked part of the map.

The cryo-EM reconstructions contain well-resolved densities for peptide group oxygen atoms (as visible in the examples shown in Extended Data Fig. 2f), which unambiguously determines the handedness of the filaments. The handedness of the filaments is constant and, therefore, not influenced by which part of the map is focussed on. To clarify this, we have added the following to the revised manuscript (line 137),

'The protein backbone, including peptide group oxygen atoms, and amino acid side chains were unambiguously resolved in our 2.8 Å cryo-EM reconstruction...'

5. L230 – does one report establish a hallmark? Perhaps 'marker' might be more appropriate?

We used the term 'hallmark' to refer to a characteristic feature. However, we are happy to replace this with the term 'marker' to avoid misinterpretation (line 232 of the revised manuscript).

6. L273-4 – is this a primer for the next paper that the authors intend to send to Nature? If they

have these data I would recommend including them here. The novelty of a second observation might preclude publication in a journal of the highest tier...

No, this is not a primer for another paper already in preparation. We do not have these data or samples. We think that this is an important hypothesis arising from our work, which will hopefully spark follow up studies.

7. L281 seems grammatically off – needs some rewording.

Thank you for spotting this typo, which we have now corrected as follows (line 292 in the revised manuscript),

'The discovery of a second protein in the filaments of FTLN-TDP Type C offers new avenues to investigate the currently enigmatic mechanisms of protein assembly in neurodegenerative diseases.'

8. L316 – is there any evidence for this? Presumably there is an extensive body of immunohistochemistry looking for coincidence of other proteins with these deposits. Is there any indication within this literature that the authors speculation is supported by evidence? It seems to me that the likelihood of such densities belonging to the fuzzy coat of the core components, with their effectively infinite local concentration, is a much more likely scenario. Indeed, there is an extensive body of research using immunohistochemistry and mass spectrometry showing coincidence of other proteins with these deposits, including references 48–50 in this manuscript. The structure presented here, which includes a peptide likely derived from the C-terminus of TDP-43, but also a second protein, ANXA11, shows that both scenarios are possible, as discussed in this paragraph. To clarify this, we have amended the paragraph (line 332 in the revised manuscript) as follows,

'Alternatively, our finding of heteromeric amyloid filaments raises the distinct possibility that these peptides may be derived from other proteins.'

9. I found the overlays in ED-Fig4a hard work – i.e. not immediately accessible – I wonder whether the authors might rework the colour scheme and/or labelling to make the figure clearer. We agree and have increased the transparency of the alternative conformations in a revised version of this figure to make it more accessible. We have also added the conformation to the

residue labels to further clarify this. We do not wish to change the colour scheme, because it relates to specific regions of TDP-43 and is consistent with previous publications.

A few queries about the data...

10. Their 2.8Å and 2.9Å maps... (I don't know why they use a different number of sig figs to describe resolution of maps – I would suggest their 2.75Å is 2.8Å and they accept the loss of nominal resolution!) come from 2.9% and 1.7% of the data. What is in the >95% of the data that is not reported on?

We have now changed the nominal resolution of the reconstruction of the main filament conformation for individual 1 to 2.8 Å as suggested.

With regards to the data that was excluded from the final reconstructions, these either A.) did not contain filaments (included erroneously during manual annotation of the micrographs) or B.) were consistent with the final filament reconstructions, but of lower resolution. 2D and 3D classification were used to remove data that did not contain filaments. In addition, 3D classification was used to remove data consistent with the final filament reconstructions but of lower resolution. We have included a figure below to illustrate this.

Figure 1: 3D classification of filament segments for individual 1.

11. And on a related note, does this 95% contain any hint of homomeric fibril (of either protein). I assume the answer is ‘no’ based on the sections about immunohistochemistry – but did they check – it’s not clear for example how good the antibodies are...

As shown in the previous response to point 10, we checked this using 3D classification and did not observe any other types of filaments in the datasets, including homomeric filaments.

12. And did they back check in previous datasets knowing the structure and helical parameters of the heterotypic fibril to see if it was in previous datasets from FTLD types A/B.

We also used the same 2D and 3D classification pipelines in the FTLD-TDP Type A and ALS/FTLD-TDP Type B studies and did not observe any filament types other than those reported in the associated papers, including heteromeric filaments.

13. How did they resolve the heterogeneity obviously in their data (obvious from the fact that less than 5% of a dataset ended up in fibril reconstructions? There is little detail, and it might benefit from a processing pipeline ED figure. For example (L482) – what is the definition of suboptimal in “remove suboptimal segments”?

Please see our response to point 10, where we explain the 2D and 3D classification pipelines used to resolve heterogeneity. For 2D classification 'suboptimal' refers to data that did not contain filaments, whereas for 3D classification it also refers to data consistent with the final filament reconstructions but of lower resolution. We have revised the manuscript to clarify this (line 790), which now reads,

'Reference-free two-dimensional (2D) classification was performed to remove segments that did not contain filaments... 3D classification was used to further remove segments that did not contain filaments, as well as to separate filament segments with alternative conformations. To achieve higher resolutions, 3D classification was also used to remove lower-resolution filament segments consistent with the final reconstructions.'

Reviewer Reports on the First Revision:

Referees' comments:

Referee #1 (Remarks to the Author):

The authors have addressed all of my minor concerns, and I believe that the revised manuscript is now ready for publication.

Referee #3 (Remarks to the Author):

I read with interest the thoughtful response of the authors to my (and other reviewers') comments. The changes made make the manuscript better, and I remain strongly supportive of publication. I would push back on a few of their responses to my questions, but I leave this to the discretion of the editors/author team. The MS certainly does not require another round of review.

Regarding Pt 4 - I did not suggest that the handedness go the filaments changes - I was suggesting that the resolution of the map is not consistent across the whole density. From the density shown, I do not think the hand is unambiguously determined - it seems borderline to this reviewer.

Regarding pt 8 - I disagree, but it's their discussion. Note that the actual manuscript that I see does not contain the word 'distinct' - which the response suggest it should. The text is better for me without it!

Regarding pt 13 - the explanation is fine, but I'd urge them to use 'remove images'? I would say that 'segments'='part of a fibril'

Author Rebuttals to First Revision:

Referees' comments:

Referee #1 (Remarks to the Author):

The authors have addressed all of my minor concerns, and I believe that the revised manuscript is now ready for publication.

Referee #3 (Remarks to the Author):

I read with interest the thoughtful response of the authors to my (and other reviewers') comments. The changes made make the manuscript better, and I remain strongly supportive of publication. I would push back on a few of their responses to my questions, but I leave this to the discretion of the editors/author team. The MS certainly does not require another round of review.

Regarding Pt 4 - I did not suggest that the handedness go the filaments changes - I was suggesting that the resolution of the map is not consistent across the whole density. From the density shown, I do not think the hand is unambiguously determined - it seems borderline to this reviewer.

We have now included larger segments of density, focussed on peptide oxygen atoms, in Extended Data Fig. 3f to better illustrate densities for peptide group oxygen atoms. It is well-established that densities for peptide group oxygen atoms allow straightforward identification of the handedness of filaments in maps with resolutions beyond 2.9 Å (for example see Lövestam *et al.* 2022. eLife 11:e76494). The resolution of our map is 2.8 Å, as shown in Extended Data Fig. 2. The map is also publicly available at the EMDB (accession code EMD-50628), as stated in the Data Availability section of the manuscript.

Regarding pt 8 - I disagree, but it's their discussion. Note that the actual manuscript that I see does not contain the word 'distinct' - which the response suggest it should. The text is better for me without it!

We maintain that the structure of heterotypic filaments of ANXA11 and TDP-43 shows that different proteins can co-assemble, raising the possibility that unidentified isolated peptide-like densities in other filament structures might be derived from different proteins. An example from the literature is a selection of structures of recombinant amyloid- β 42 filaments (EMD-21501, EMD-29037, EMD-28741). These contain isolated peptide-like density islands that cannot be accounted for by the protein chains that form the rest of the filament core folds. As these filaments were assembled *in vitro*, these peptide-like densities are presumably derived from different amyloid- β 42 molecules, rather than a distinct protein. In order to clarify this point, we have amended the paragraph in question (line 344 in the revised manuscript) as follows,

'...They may derive from the same protein that makes up the rest of the filament fold, similar to the isolated peptide associated with TDP-43 residues F289–R293 described here. Alternatively, our finding of heteromeric amyloid filaments raises the possibility that these peptides may be derived from different proteins...'

Regarding pt 13 - the explanation is fine, but I'd urge them to use 'remove images'? I would say that 'segments'='part of a fibril'

We agree that our use of the word 'segments' to refer to both regions of images and filaments has the potential to cause confusion. We have therefore replaced the word 'segments' with the word 'image coordinates' when referring to images and retain the use of the word 'segments' to refer to filaments only. The sentence in question (line 820 in the revised manuscript) now reads,

'Reference-free two-dimensional (2D) classification was performed to remove image coordinates that did not contain filaments... 3D classification was used to further remove image coordinates that did not contain filaments, as well as to separate filament segments with alternative conformations. To achieve higher resolutions, 3D classification was also used to remove lower-resolution filament segments consistent with the final reconstructions.'